# Skill Expansion and Composition in Parameter Space

**Tenglong Liu**[1][*]**, Jianxiong Li**[2][*]**, Yinan Zheng**[2]**, Haoyi Niu**[2]**, Yixing Lan**[1][†]**, Xin Xu**[1][†]**,
Xianyuan Zhan**[2,3,4][†]
[1] National University of Defense Technology, [2] Tsinghua University,
[3] Shanghai Artificial Intelligence Laboratory, [4] Beijing Academy of Artificial Intelligence
`ltl@nudt.edu.cn,li-jx21@mails.tsinghua.edu.cn,`
`{xinxu,lanyixing16}@nudt.edu.cn,zhanxianyuan@air.tsinghua.edu.cn`

## Abstract

Humans excel at reusing prior knowledge to address new challenges and developing skills while solving problems. This paradigm becomes increasingly popular in the development of autonomous agents, as it develops systems that can self-evolve in response to new challenges like human beings. However, previous methods suffer from limited training efficiency when expanding new skills and fail to fully leverage prior knowledge to facilitate new task learning. In this paper, we propose Parametric Skill Expansion and Composition (PSEC), a new framework designed to iteratively evolve the agents' capabilities and efficiently address new challenges by maintaining a manageable skill library. This library can progressively integrate skill primitives as plug-and-play Low-Rank Adaptation (LoRA) modules in parameter-efficient finetuning, facilitating efficient and flexible skill expansion. This structure also enables the direct skill compositions in parameter space by merging LoRA modules that encode different skills, leveraging shared information across skills to effectively program new skills. Based on this, we propose a context-aware module to dynamically activate different skills to collaboratively handle new tasks. Empowering diverse applications including multi-objective composition, dynamics shift, and continual policy shift, the results on D4RL, DSRL benchmarks, and the DeepMind Control Suite show that PSEC exhibits superior capacity to leverage prior knowledge to efficiently tackle new challenges, as well as expand its skill libraries to evolve the capabilities. Project website: https://ltlhuuu.github.io/PSEC/.

## 1 Introduction

Humans excel at using existing skills and knowledge to tackle new tasks efficiently, while continually evolving their capabilities to rapidly adapt to new tasks. (Driscoll et al., 2024; Courellis et al., 2024; Eppe et al., 2022; Eichenbaum, 2017). This fundamental approach to problem-solving highlights a key aspect of human intelligence that is equally crucial for autonomous agents. However, most current decision-making algorithms adhere to a *tabula rasa* paradigm, where they are trained from scratch without utilizing any prior knowledge or resources (Akkaya et al., 2019; Berner et al., 2019; Silver et al., 2016), leading to severe sample inefficiency and elevated cost when the agent encounters new tasks (Agarwal et al., 2022; Peng et al., 2019; Du & Kaelbling, 2024). Therefore, in this paper, we aim to explore the capability of autonomous agents to leverage and expand upon their existing knowledge base in novel situations to enhance learning efficiency and adaptability.

While some existing studies, such as continual learning (Liu et al., 2024a; Gai & Wang, 2024), compositional policies (Peng et al., 2019; Janner et al., 2022; Ajay et al., 2023), or finetuning-based methods (Agarwal et al., 2022), aim to replicate this process, they jointly failed to tackle several key limitations. *1) Catastrophic forgetting*: these approaches typically lack a fundamental mechanism to guarantee continuous improvement when acquiring new skills, making the autonomous agents very susceptible to overfitting on new tasks while forgetting previously learned skills without proper regularization (Liu et al., 2023c; 2024a; Gai & Wang, 2024); *2) Limited efficiency in learning new tasks*: Some methods avoid the catastrophic forgetting problem by adopting a parameter-isolation

---

[*]Equal contribution.
[†]Corresponding Authors.

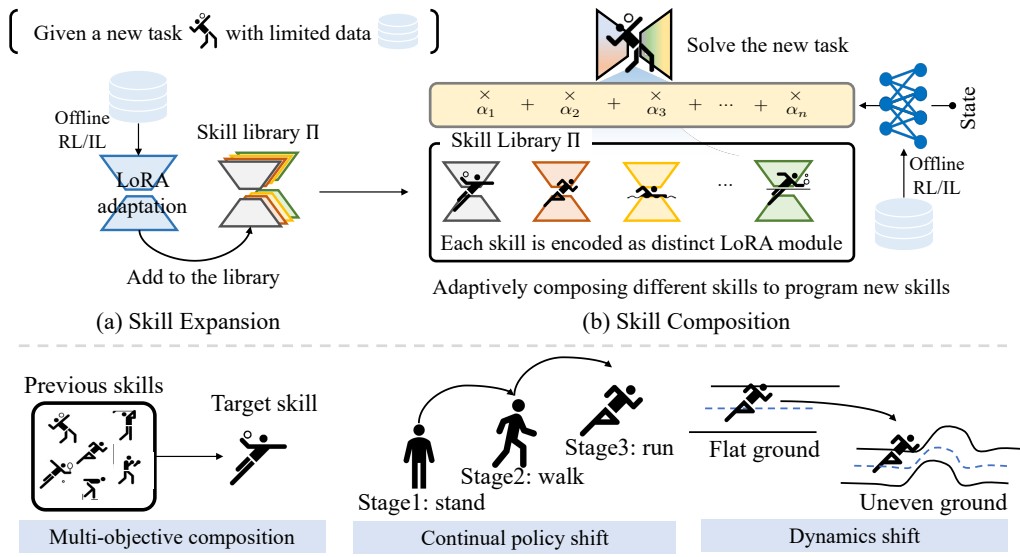

(a) Skill Expansion

(b) Skill Composition

(c) Diverse applications of PSEC framework

Figure 1: PSEC framework and its application in diverse scenarios. (a) We maintain a skill library that contains many skills primitives and can progressively expand by adding new LoRA modules. (b) Then we train a context-aware compositional network to adaptively compose different elements in the skill library to solve new tasks. (c) PSEC framework is versatile to diverse applications where reusing prior knowledge is crucial.

approach via encoding new skills in independent new parameters. However, they typically do not fully utilize prior knowledge from old skills to enhance training in current tasks, lacking an efficient way to learn new skills in terms of both parameters and training samples (Peng et al., 2019; Zhang et al., 2023a), leading to tremendous costs as the number of skills progressively grows.

In order to deal with the above problems, we propose Parametric Skill Expansion and Composition (PSEC), a framework that facilitates efficient self-evolution of autonomous agents by maintaining a skill library that progressively integrates new skills, facilitating rapid adaptation to evolving demands. The key mechanism of PSEC is to utilize the primitives in the skill library to tackle new challenges by exploiting the shared information across different skills within the parameter space. As shown in Figure 1 (a), we adopt the Low-Rank Adaptation (LoRA) (Hu et al., 2021) approach, which encodes skills as trainable parameters injected into existing frozen layers. This parameter-isolation approach naturally resolves the catastrophic forgetting problem, and significantly reduces computational burden due to the low-rank decomposition structure. This efficient modular design allows for managing skills as plug-and-play modules, and thus can directly blend different abilities within the parameter space to interpolate new skills (Clark et al., 2024), as shown in Figure 1 (b). The proposed PSEC approach can leverage more shared or complementary structures across skills for optimal compositions. Based on this insight, a context aware module is designed to adaptively compose skills and each primitive is modeled by diffusion models to ensure both flexibility and expressiveness in composition. Through iterative expansion and composition, PSEC can continually evolve and efficiently tackle new tasks, offering one promising pathway for developing human-level autonomous agents.

Empowering diverse settings including multi-objective composition, continual policy shift and dynamics shift, PSEC demonstrates its capacity to evolve and effectively solve new tasks by leveraging prior knowledge, evaluated on the D4RL (Fu et al., 2020), DSRL (Liu et al., 2023a) and DeepMind Control Suite (Tassa et al., 2018), showcasing significant potential for real-world applications.

## 2 RELATED WORKS

**Compositional Policies**. Some previous methods try to leverage prior knowledge relying on pre-trained primitive policies. More specifically, these methods used compositional networks in a hierarchical structure to adaptively compose primitives to form complex behaviors (Peng et al., 2019; Qureshi et al., 2020; Pertsch et al., 2021; Merel et al., 2019; 2020). However, their expressiveness is limited by the expressiveness of simple Gaussian primitives. Recently, due to the strong expressiveness of the diffusion models and its inherent connection with Energy-Based Models (LeCun et al.,

2006), many compositional policies have been approached by diffusion model. Diffusion models learn the gradient fields of an implicit energy function, which can be combined at inference time to generalize to new complex distribution readily (Janner et al., 2022; Wang et al., 2024b; Du & Kaelbling, 2024; Liu et al., 2022; Luo et al., 2024b). However, these approaches rely on independently trained policies with fixed combination weights, which lack the flexibility to adapt to complex scenarios. Moreover, most previous methods can only combine skills after the policy distribution generation of each skill. Therefore, they fail to fully utilize the shared features of different skills to achieve optimal compositions. We systematically investigate the advantages of skill composition within the parameter space, and compose skills in a context-aware manner with each skill modeled as a diffusion model. This ensures both flexibility and expressiveness in composing complex behaviors.

**Continual Learning for Decision Making**. Current continual learning methods for decision making, including continual reinforcement learning (RL) and imitation learning (IL), primarily focus on mitigating catastrophic forgetting of prior knowledge when learning new tasks. They can be roughly classified into three categories: structure-based (Smith et al., 2023; Wang et al., 2024d), regularization-based (Kessler et al., 2020), and rehearsal-based methods (Liu et al., 2024a; Peng et al., 2023). Different from previous continual RL and IL approaches, our study focuses on leveraging existing skills to facilitate efficient new task learning and enables the extension of skill sets. In addition, it naturally solves the catastrophic forgetting challenge due to the parameter isolation induced by the LoRA module (Liu et al., 2023c), directly bypassing the key challenges of existing continual learning methods.

## 3 METHODS

We propose PSEC, a generic framework that can efficiently reuse prior knowledge and self-evolve to address emerging new tasks. Next, we will elaborate on our problem setup and technical details.

### 3.1 PRELIMINARY

**Diffusion Model for Policy Modeling**. Recently, diffusion models have become popular for policy modeling because of their superior expressiveness to model complex distributions (Wang et al., 2023; Chen et al., 2022; Lu et al., 2023; Zheng et al., 2025). Considering a policy distribution $\pi(a|s)$ and a sample $(s, a)$ drawn from an empirical dataset $\mathcal{D}$ of $\pi(a|s)$, the diffusion process (Ho et al., 2020) progressively introduces Gaussian noise to the sample over $T$ steps, producing a sequence of noisy samples $a_0, a_1, ..., a_T$ with $a_0 = a$ following the forward Gaussian kernel:

$$q(a_t|a_{t-1}) = \mathcal{N}(a_t; \sqrt{1-\beta_t}a_{t-1}, \beta_t I), \quad q(a_t|a_0) = \mathcal{N}(a_t; \sqrt{\bar{\rho}_t}a_0, (1-\bar{\rho}_t)I), \quad (1)$$

where $\rho_t := 1 - \beta_t, \bar{\rho}_t = \prod_{t=1}^{t} \rho_t$, and the noise is controlled by a variance schedule $\beta_1, ..., \beta_t$ to ensure $p(a_T) = \mathcal{N}(0, I)$. The denoise process aims to recover the sample from $p(a_T)$ by learning a conditional distribution $p_\theta(a_{t-1}|a_t, s)$. The policy $\pi_\theta(a|s)$ is typically modeled as:

$$\pi_\theta(a|s) = p(a_t) \prod_{t=1}^{T} p_\theta(a_{t-1}|a_t, s); p_\theta(a_{t-1}|a_t, s) = \mathcal{N}(a_{t-1}; \mu_\theta(a_t, t, s), \Sigma_\theta(a_t, t, s)), \quad (2)$$

where $\Sigma_\theta = \beta_t I$ is set as untrained time-dependent constants and $\mu_\theta(a_t, t, s) = \frac{1}{\sqrt{\rho_t}}(a_t - \frac{\beta_t}{\sqrt{1-\bar{\rho}_t}}\epsilon_\theta(a_t, t, s))$ is reparameterized by $\epsilon_\theta$. The trainable parameter $\theta$, modeled by deep networks, can be optimized via minimizing the following objective by predicting the noise:

$$\mathcal{L}_{\text{diff}}(\theta) = \mathbb{E}_{t \sim \mathcal{U}, \epsilon \sim \mathcal{N}(0,I), (s,a) \sim \mathcal{D}} \left[ w(s, a) \left\| \epsilon - \epsilon_\theta \left( \sqrt{\bar{\rho}_t}a + \sqrt{1-\bar{\rho}_t}\epsilon, t, s \right) \right\|^2 \right]. \quad (3)$$

where $\mathcal{U}$ is uniform distribution over the discrete set $\{1, ..., T\}$. $w(s, a)$ is a flexible weight function that encodes human preference (Zheng et al., 2024). For example, $w(s, a) \propto f(A(s, a)), f \geq 0$ with $A(s, a)$ as the advantage function leads to weighted behavior cloning (BC) in offline reinforcement learning (RL) (Zheng et al., 2024; Kostrikov et al., 2022; Xu et al., 2023), and $w(s, a) := 1$ degenerates to traditional BC (Chen et al., 2023). After obtaining the approximated $\mu_\theta$ and $\Sigma_\theta$, we can substitute them into Eq. (2) to iteratively denoise and obtain actions conditioned on the state.

**Problem Setups**. We consider a Markov Decision Process with $s \in \mathcal{S}$ and $a \in \mathcal{A}$ are state and action space, $\mathcal{P} : \mathcal{S} \times \mathcal{A} \to \Delta(\mathcal{S})$ is transition dynamics, and $r : \mathcal{S} \times \mathcal{A} \to \mathbb{R}$ is reward function. We assume

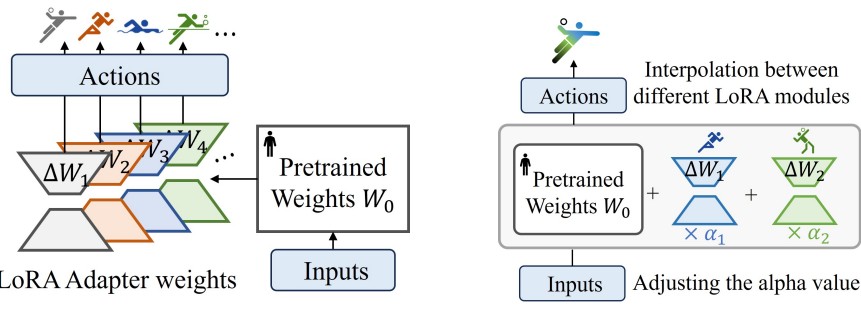

(a) Learning new skills using LoRA modules.      (b) Interpolation in LoRA modules.

Figure 2: (a) Each skill is encoded in separate LoRA modules respectively. (b) By adjusting the composing weights $\alpha_i$, different LoRA modules can merge together to interpolate new skills.

the state space $\mathcal{S}$ and action space $\mathcal{A}$ remain unchanged during training, which is a mild assumption in many relevant works (Peng et al., 2019; Ajay et al., 2023; Nair et al., 2020; Liu et al., 2024b; Dai et al., 2024). We consider an agent with $\pi_0$ as its initial policy and then is progressively tasked with new tasks $\mathcal{T}_i, i = 1, 2, ...$, with differences in the rewards $r$ or dynamics $\mathcal{P}$, to mirror real-world scenarios with non-stationary dynamics or new challenges continually emerge (Luo et al., 2024a). Each task is provided with several expert demonstrations $\mathcal{D}_e^{\mathcal{T}_i} := \{(s, a)\}$ or a mixed-quality dataset with reward labels $\mathcal{D}_o^{\mathcal{T}_i} := \{(s, a, r_i, s')\}$. So, we can use either offline RL or imitation learning (IL) (Gong et al., 2024b) to adapt to the new challenges (Liu et al., 2024b). Inspired by previous works (Peng et al., 2019; Barreto et al., 2018; Zhang et al., 2023a), we maintain a policy library $\Pi$ to store the policies associated with different tasks and aim to utilize the prior knowledge to enable efficient policy learning and gradually expand it to incorporate new abilities across training.

$$\Pi = \{\pi_0, \pi_1, \pi_2, \pi_3, ...\}. \tag{4}$$

We aim to explore *1) Efficient Expansion*: How to manage the skill library $\Pi$ to learn new skills in an efficient and manageable way, and *2) Efficient Composition*: How to fully utilize the prior knowledge from primitives in the skill set $\Pi$ to tackle the emerging challenges.

## 3.2 EFFICIENT POLICY EXPANSION VIA LOW-RANK ADAPTATION

For the first objective, previous methods typically train each primitive from scratch in a tabula rasa paradigm (Peng et al., 2019; Janner et al., 2022; Lu et al., 2023), failed to leverage the prior knowledge in $\Pi$ to efficiently obtain a good skill primitive. This presents significant issues in terms of computational efficiency when the number of skills grows. To mitigate these challenges, we turn to Parameter-Efficient Fine-Tuning (PEFT) (Ding et al., 2023), which has proven highly effective in various natural language processing and computer vision applications. One of the most popular PEFT implementations is LoRA (Hu et al., 2022). It injects trainable low-rank decomposed matrices into the pretrained layer to avoid overfitting with limited adaptation data and significantly reduces computational and memory burden. Inspired by this, we try to employ LoRA to efficiently learn new skills given solely limited data for the target skill.

**Policy Expansion via Low-Rank Adaptation**. We consider a pretrained policy $\pi_0$ and denote $W_0 \in \mathbb{R}^{d_{\text{in}} \times d_{\text{out}}}$ as its associated weight matrix. Directly finetuning $W_0$ to adapt to new skills might be extremely inefficient (Liu et al., 2023c), instead, we introduce a tune-able LoRA module $\Delta W$ upon $W_0$, *i.e.*, $W_0 + \Delta W = W_0 + BA$ to do the adaptation and keep $W_0$ frozen, where $B \in \mathbb{R}^{d_{\text{in}} \times n}$, $A \in \mathbb{R}^{n \times d_{\text{out}}}$ and $n \ll \min(d_{\text{in}}, d_{\text{out}})$. Specifically, the input feature of the linear layer is denoted as $h_{\text{in}} \in \mathbb{R}^{d_{\text{in}}}$, and the output feature of the linear layer is $h_{\text{out}} \in \mathbb{R}^{d_{\text{out}}}$, the final output of a LoRA augmented layer can be calculated through the following forward process:

$$h_{\text{out}} = (W_0 + \alpha \Delta W)h_{\text{in}} = (W_0 + \alpha BA)h_{\text{in}} = W_0 h_{\text{in}} + \alpha BA h_{\text{in}}, \tag{5}$$

where $\alpha$ is a weight to balance the pre-trained model and LoRA modules. This operation naturally prevents catastrophic forgetting in a parameter isolation approach, and the low-rank decomposition structure of $A$ and $B$ significantly reduces the computational burden. Benefiting from this lightweight characteristic, we can manage numerous LoRA modules $\{\Delta W_i = B_i A_i | i \in 1, 2, ..., k\}$ to encode different skill primitives $\pi_i$, respectively, as shown in Figure 2a. This flexible approach allows us to easily integrate new skills based on existing knowledge, while also facilitating library

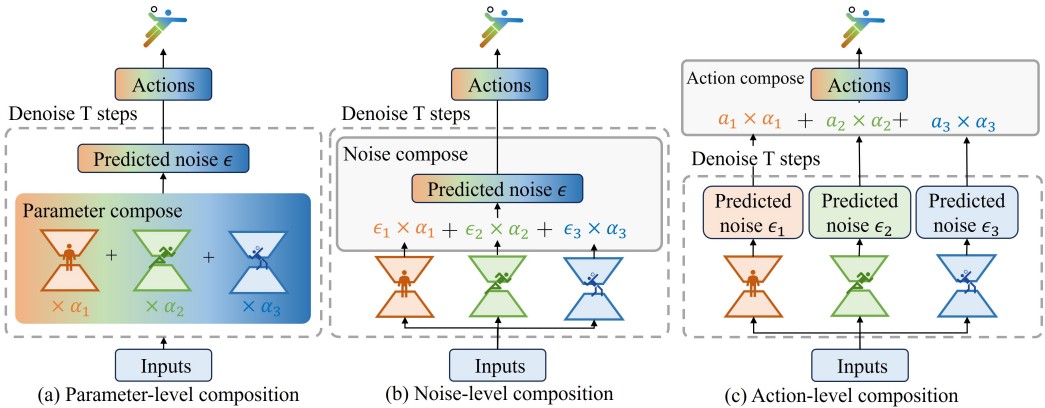

Figure 3: Comparison between parameter-, noise-, and action-level composition. Parameter-level composition offers more flexibility to leverage the shared or complementary structure across skills to compose new skills. Noise- and action-level composition, however, is too late to benefit from this information.

management by removing suboptimal primitives and retaining the effective ones. More importantly, by adjusting the value of $\alpha$, it holds the potential to interpolate the pretrained skill in $W_0$ and other primitives in $\Delta W_i$ (Clark et al., 2024) to generate novel skills, as shown in Eq. (6) and Figure 2b.

$$W = W_0 + \sum_{i=1}^{k} \alpha_i \Delta W_i = W_0 + \sum_{i=1}^{k} \alpha_i B_i A_i, \tag{6}$$

where $\alpha_i$ is the weight to interpolate pre-trained weights and LoRA modules. This interpolation property has been explored in fields like text-to-image generation (Clark et al., 2024) and language modeling (Zhang et al., 2023b), but its application in decision-making scenarios remains highly underexplored, despite LoRA has proven efficacy in skill acquisition (Liu et al., 2023c). Next, we will elaborate on how to effectively combine LoRA modules to adapt to decision-making applications.

### 3.3 CONTEXT-AWARE COMPOSITION IN PARAMETER SPACE

Effectively combining skills encoded as different LoRA modules to solve new tasks is crucial. Previous methods (Du & Kaelbling, 2024; Ajay et al., 2023; Janner et al., 2022) typically rely on fixed combinations of skills, resulting in limited compositional flexibility. This approach may be acceptable in static domains like language models, but it falls short in decision-making applications where dynamic skill composition is crucial. For example, in autonomous driving, the ability to dynamically prioritize skills of obstacle avoidance in potential collision scenarios, or acceleration when speeds are suboptimal, is essential. Naively adopting a fixed set of $\alpha_i$ like previous approaches (Du & Kaelbling, 2024; Ajay et al., 2023; Janner et al., 2022; Clark et al., 2024), however, cannot adequately support such flexible deployment of skills based on real-time environmental demands.

**Context-aware Composition**. We propose a simple yet effective context-aware composition method that adaptively leverages pretrained knowledge to optimally address the encountering tasks according to the agent's current context. Specifically, we introduce a context-aware modular $\alpha(s; \theta) \in \mathbb{R}^k$ with $\alpha_i$ as its $i$-th dimension. The composition method can be expressed by Eq. (7):

$$W(\theta) = W_0 + \sum_{i=1}^{k} \alpha_i(s; \theta) \Delta W_i = W_0 + \sum_{i=1}^{k} \alpha_i(s; \theta) B_i A_i. \tag{7}$$

Here, $\alpha(s; \theta)$ adaptively adjusts output weights based on the agent's current situation $s$ with the parameter $\theta$ optimized via minimizing the diffusion loss in Eq. (3). Note that the trainable parameter $\theta$ lies solely in the composition network $\alpha_\theta$ with the pretrained weights $W_0$ and all LoRA modules $\Delta W_i$ being kept frozen, thus $\theta$ can be efficiently trained in terms of both samples and parameters.

**Parameter-level *v.s.* Action-level Composition**. Careful readers may notice that our context-aware composition is similar to previous works that adaptively compose Gaussian primitive skills to create complex behaviors (Peng et al., 2019; Qureshi et al., 2020), such as the one shown in Eq. (8) (Peng et al., 2019):

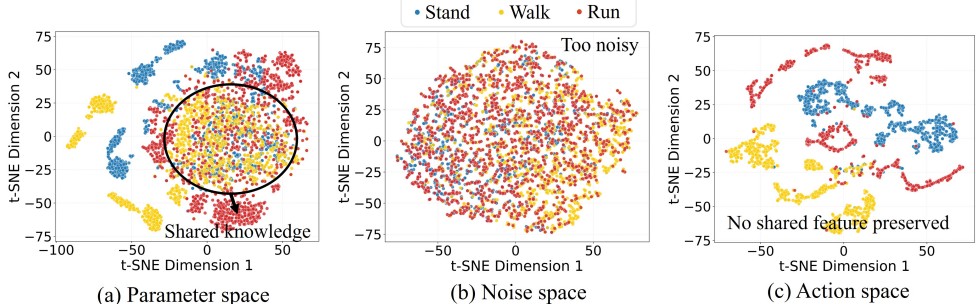

Figure 4: t-SNE projections of samples from different skills in parameter, noise, and action space. The parameter space exhibits a good structure for skill composition, where skills share common knowledge while retaining their unique features to avoid confusion. Noise and action spaces are either too noisy to clearly distinguish between skills or fail to capture the shared structure across them. See Appendix C.4 for details.

$$\pi(a|s) = \frac{1}{Z(s)} \prod_{i=1}^{k} \pi_i(a|s)^{\alpha_i(s;\theta)}, \quad \pi_i(a|s) = \mathcal{N}\left(\mu_i(s), \Sigma_i(s)\right),$$ (8)

where $\alpha(s;\theta)$ is optimized to combine the policy distributions $\pi_i, i = 0, ..., k$ to collaboratively build a new policy distribution $\pi$ to solve the new task.

However, these two methods differ fundamentally in their stages of composition, mirroring the advantages of *early fusion* over *late fusion* across various domains (Gadzicki et al., 2020; Wang et al., 2024e). PSEC employs a *parameter-level composition*, where different skills are seamlessly integrated within the parameter space. By contrast, Eq. (8) represents an *action-level composition* that explicitly combines the output distributions of various skills. In comparison, *parameter-level composition* will be more efficient, as it can leverage more shared or complementary information between different skills to enhance compositionality and overall performance before generating the final policy distribution (Shazeer et al., 2016; Wang et al., 2024d). Conversely, *action-level composition* only merges skills after the action generation, which is too late to effectively leverage features across skills for optimal composition. Besides, previous *action-level* methods typically employ simple Gaussian primitives to construct their skill library, significantly limiting its expressiveness.

**Parameter-level *v.s.* Noise-level Composition**. Some approaches use diffusion models for policy modeling and exhibit remarkable compositionality by identifying its connections to Energy-Based Models (Du & Kaelbling, 2024; Wang et al., 2024b; Janner et al., 2022; Ajay et al., 2023; Lu et al., 2023). Specifically, the noise predicted by diffusion models can be regarded as the gradient field of some energy functions. It thus can be directly merged to form new skills during sampling in a *noise-level composition*, as shown in Eq. (9). This is equivalent to doing a logical operation on the energy functions to form complex behaviors (Du et al., 2023; Liu et al., 2022; LeCun et al., 2006).

$$\epsilon(a_t, t, s) = \sum_{i=0}^{k} \alpha_i \epsilon_i(a_t, t, s).$$ (9)

Here, $\epsilon_i$ represents the predicted noise derived from various skills, while $\epsilon$ is the aggregated noise resulting from their composition. Utilizing $\epsilon$ for denoising in Eq. (2) allows for the generation of a joint distribution of skills, thereby facilitating the effective composition of these diverse capabilities (Ajay et al., 2023; 2024; Janner et al., 2022). However, these methods employ fixed weights $\alpha_i$ for policy composition, limiting their flexibility and adaptability in dynamical scenarios where real-time adjustment on the compositional weights is required. In our paper, PSEC not only employs diffusion models to enhance the expressiveness of primitives, but also adaptively adjusts the context-aware compositional weights to enhance compositional flexibility. Additionally, this *noise-level composition* also tends to be less effective than *parameter-level composition*, as the latter integrates different skills at an earlier stage, leading to improved performance, as shown in Figure 3.

**Empirical Observations**. To evaluate the advantages of parameter-level composition over other levels of composition, we employ t-SNE (Van der Maaten & Hinton, 2008) to project the output features of LoRA modules into a 2D space, alongside the noise and generated actions of various skills. Figure 4 illustrates that in the parameter space, different skills not only share common knowledge, but also retain their unique features to avoid confusion. In contrast, noise and action spaces

Table 1: Normalized DSRL (Liu et al., 2023a) benchmark results. Costs below 1 indicates safety. ↑: the higher the better. ↓: the lower the better. Results are averaged over 20 evaluation episodes and 4 seeds. **Bold**: Safe agents with costs below 1. **Blue**: Safe agents achieving the highest reward.

| Task | BC | | CDT | | CPQ | | COptiDICE | | FISOR | | ASEC | | NSEC | | PSEC | |
|---|---|---|---|---|---|---|---|---|---|---|---|---|---|---|---|---|
| | reward ↑ | cost ↓ | reward ↑ | cost ↓ | reward ↑ | cost ↓ | reward ↑ | cost ↓ | reward ↑ | cost ↓ | reward ↑ | cost ↓ | reward ↑ | cost ↓ | reward ↑ | cost ↓ |
| easysparse | 0.32 | 4.73 | **0.05** | **0.10** | -0.06 | 0.24 | 0.94 | 18.21 | **0.38** | **0.53** | 0.95 | 5.8 | **0.55** | **0.08** | **0.55** | **0.02** |
| easymean | 0.22 | 2.68 | **0.27** | **0.24** | -0.06 | 0.24 | 0.74 | 14.81 | **0.38** | **0.25** | **0.63** | **0.75** | 0.39 | 0.54 | 0.37 | 0.00 |
| easydense | 0.20 | 1.70 | 0.43 | 2.31 | **-0.06** | **0.29** | 0.60 | 11.27 | **0.36** | **0.25** | 0.85 | 5.28 | 0.76 | 1.45 | **0.51** | **0.01** |
| mediumsparse | 0.53 | 1.74 | 0.26 | 2.20 | **-0.08** | **0.18** | 0.64 | 7.26 | **0.42** | **0.22** | 0.93 | 2.52 | **0.60** | **0.08** | **0.76** | **0.03** |
| mediummean | 0.66 | 2.94 | 0.28 | 2.13 | **-0.08** | **0.28** | 0.73 | 8.35 | **0.39** | **0.08** | 0.74 | 1.00 | 0.82 | 2.87 | **0.61** | **0.01** |
| mediumdense | 0.65 | 3.79 | **0.29** | **0.77** | -0.08 | 0.20 | 0.91 | 9.52 | **0.49** | **0.44** | **0.81** | **0.52** | 0.76 | 0.27 | 0.66 | 0.02 |
| hardsparse | 0.28 | 1.98 | **0.17** | **0.47** | -0.04 | 0.28 | 0.34 | 7.34 | **0.30** | **0.01** | 0.30 | 0.41 | 0.34 | 1.21 | **0.34** | **0.04** |
| hardmean | 0.34 | 3.76 | 0.28 | 3.32 | -0.05 | 0.24 | 0.36 | 7.51 | **0.26** | **0.09** | 0.46 | 1.05 | **0.38** | **0.32** | **0.39** | **0.07** |
| harddense | 0.40 | 5.57 | 0.24 | 1.49 | -0.04 | 0.24 | 0.42 | 8.11 | **0.30** | **0.34** | **0.36** | **0.82** | 0.19 | 0.03 | 0.34 | 0.07 |
| **MetaDrive Average** | 0.40 | 3.21 | 0.25 | 1.45 | **-0.06** | **0.24** | 0.63 | 10.26 | **0.36** | **0.25** | 0.67 | 2.02 | **0.53** | **0.76** | **0.50** | **0.03** |
| AntRun | 0.73 | 11.73 | 0.70 | 1.88 | **0.00** | **0.00** | 0.62 | 3.64 | **0.45** | **0.03** | 0.74 | 4.97 | 0.79 | 6.81 | **0.59** | **0.33** |
| BallRun | 0.67 | 11.38 | **0.32** | **0.45** | 0.85 | 13.67 | 0.55 | 11.32 | **0.18** | **0.00** | 0.35 | 4.35 | 0.58 | 7.46 | **0.15** | **0.95** |
| CarRun | 0.96 | 1.88 | 0.99 | 1.10 | 1.06 | 10.49 | **0.92** | **0.00** | 0.73 | 0.14 | **0.93** | **0.39** | 0.93 | 0.66 | 0.83 | 0.00 |
| DroneRun | 0.55 | 5.21 | **0.58** | **0.30** | 0.02 | 7.95 | 0.72 | 13.77 | **0.30** | **0.55** | 0.57 | 2.29 | 0.62 | 7.3 | **0.47** | **0.87** |
| AntCircle | 0.65 | 19.45 | 0.48 | 7.44 | **0.00** | **0.00** | 0.18 | 13.41 | **0.20** | **0.00** | 0.46 | 5.55 | 0.36 | 2.08 | **0.20** | **0.00** |
| BallCircle | 0.72 | 10.02 | 0.68 | 2.10 | 0.40 | 4.37 | 0.70 | 9.06 | **0.34** | **0.00** | 0.54 | 1.58 | 0.58 | 2.08 | **0.34** | **0.22** |
| CarCircle | 0.65 | 11.16 | 0.71 | 2.19 | 0.49 | 4.48 | 0.44 | 7.73 | **0.40** | **0.11** | 0.41 | 2.86 | 0.40 | 2.62 | **0.36** | **0.20** |
| DroneCircle | 0.82 | 13.78 | 0.55 | 1.29 | -0.27 | 1.29 | 0.24 | 2.19 | **0.48** | **0.00** | 0.65 | 3.60 | 0.71 | 4.93 | **0.33** | **0.07** |
| **BulletGym Average** | 0.72 | 10.58 | 0.63 | 2.09 | 0.32 | 5.28 | 0.55 | 7.64 | **0.39** | **0.10** | 0.58 | 3.20 | 0.62 | 4.24 | **0.41** | **0.33** |

are either too noisy to clearly distinguish between skills or fail to capture the shared structure across them, making the compositions in noise and action space less effective than the parameter space.

## 4 EXPERIMENTS

PSEC enjoys remarkable versatility across various scenarios since many problems can be resolved by reusing pre-trained policies and gradually evolving its capabilities during training. Thus, we present a comprehensive evaluation across diverse scenarios, including multi-objective composition, policy learning under policy shifts and dynamics shifts, to answer the following questions:

- Can the context-aware modular effectively compose different skills?
- Can our parameter-level composition outperform noise- and action-level compositions?
- Can the introduction of LoRA modules enhance training and sample efficiency?
- Can PSEC framework iteratively evolve after incorporating more skills?

### 4.1 MULTI-OBJECTIVE COMPOSITION

In many real-world applications, a complex task can be decomposed into simpler objectives, where collaboratively combining these atomic skills can tackle the complex task. In this setting, we aim to evaluate the advantages of parameter-level composition over other levels of composition in Figure 3, and the effectiveness of the context-aware modular. We consider one practical multi-objective composition scenario within the safe offline RL domain (Zheng et al., 2024). This setting requires solving a constrained MDP (Altman, 2021) to tackle a complex trilogy objective: avoiding distributional shift, maximizing rewards, and meanwhile minimizing costs. These objectives can conflict, thus requiring a nuanced composition to optimize performance effectively (Zheng et al., 2024).

**Setup**. We evaluate on a popular safe offline RL benchmark, DSRL (Liu et al., 2023a). We set $w(s, a) = 1$ in Eq. (3) to train our initial policy $\pi_0$ as a behavior policy. Then, we set $w(s, a) = \exp(A_r^*(s, a))$ and $w(s, a) = \exp(-A_h^*(s, a))$ with $A_r^*(s, a)$ and $A_h^*(s, a)$ are the optimal reward and feasible value function learned by expectile regression (Zheng et al., 2024) to train $\pi_1$ and $\pi_2$ that separately consider reward and safety performance respectively. During composition, we adopt a few filtered near-expert demonstrations that jointly consider the trilogy objective, which is too

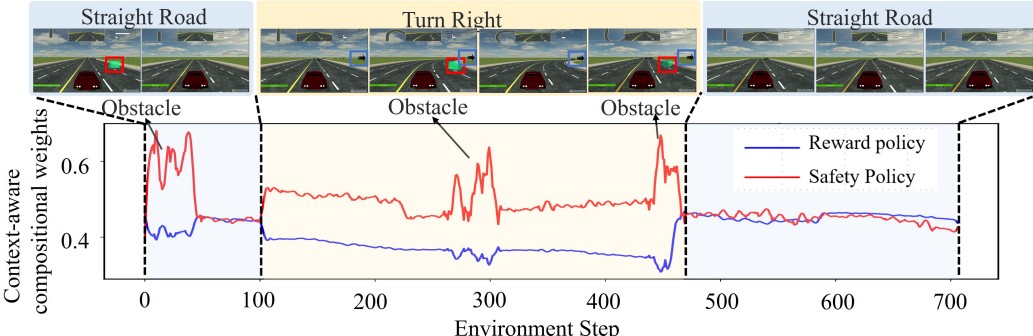

Figure 5: Output weights of context-aware modular evaluated on the MetaDrive-easymean task. The network dynamically adjusts the weights to handle real-time demands: It prioritizes safety policies when the vehicle approaches obstacles or navigates a turn while avoiding boundary lines. When there are no obstacles and the task is simply to drive straight, the focus shifts to maximizing rewards and maintaining some safety insurance.

limited to imitate good policies. However, we can adopt these data to train a context-aware modular $\alpha(s;\theta)$ in Eq. (7) to adaptively compose $\pi_{0,1,2}$ to handle the conflicts in an efficient way.

**Baselines.** To demonstrate the effectiveness of the composition in parameter space, we compare two other composition methods: *noise-level* and *action-level* composition. We denote them as NESC and ASEC respectively, where we control the only differences to PSEC being the composition stage as shown in Figure 3 to ensure a fair comparison. We also compare recent state-of-the-art (SOTA) safe offline RL methods including FISOR (Zheng et al., 2024), CDT (Liu et al., 2023b), COptiDICE (Lee et al., 2022a), CPQ (Xu et al., 2022) and BC. These traditional safe offline RL methods typically use human-tuned trade-offs to balance the trilogy objective, which is equivalent to using fixed composition weights compared to PSEC. All policies are trained on the full DSRL dataset to ensure a fair comparison (see Appendix C.1 for details).

**Main Results.** Table 1 shows that PSEC achieves a good balance between high returns and satisfactory safety performance, and simultaneously mitigates distributional shift across all tasks, enjoying highly competitive performance. In contrast, NSEC and ASEC exhibit skewed learning behaviors, where both of them fail to discover an effective composition to ensure both good safety performance and high returns, resulting in relatively poor safety outcomes despite high rewards. PSEC also outperforms all traditional safe offline RL baselines, demonstrating the necessity of context-aware composition over fixed composition when the task requires intricate balance between different elements. To further support this, we visualize the outputs of our context-aware modular $\alpha(s;\theta)$ to illustrate its adaptive capabilities. Figure 5 demonstrates that the network dynamically adjusts the weightings to combine different skills, enabling a collaborative response to real-time environmental changes. This adaptive behavior highlights the importance of dynamically adjusting the compositional weights rather than relying on a fixed combination of different skills to jointly solve a new task like previous methods (Ajay et al., 2023; Zheng et al., 2024; Janner et al., 2022).

## 4.2 CONTINUAL POLICY SHIFT SETTING

We evaluate another practical scenario where the agent is progressively tasked with new tasks. We aim to continuously expand the skill libraries to test if the capabilities of agents to learn new skills can be gradually enhanced as prior knowledge grows and test the efficiency of LoRA.

**Setup.** We conduct experiments on the DeepMind Control Suite (DMC) (Tassa et al., 2018) environments, where an agent is progressively required to stand, walk, and run. We investigate whether PSEC can leverage the standing skill to rapidly learn to walk, and then effectively combine standing and walking skills to adapt to running. For this purpose, we pretrain $\pi_0$ to learn the basic standing skill by setting $w(s,a) := 1$ in Eq. (3) trained on a expert dataset $\mathcal{D}_e^{\mathcal{T}_0}$. Subsequently, we provide small expert datasets $\mathcal{D}_e^{\mathcal{T}_1}$ for walk and $\mathcal{D}_e^{\mathcal{T}_2}$ for run, while maintaining $w(s,a) := 1$ to adapt to $\pi_1$ and $\pi_2$. After training $\pi_1$, we integrate it into the skill library $\Pi$ to assist $\pi_2$ training alongside $\pi_0$. See Appendix C.2 for detailed experimental setups.

**Baselines.** *1)* We compare NSEC and ASEC to further demonstrate the superiority of parameter- over noise- and action-level composition. *2)* We evaluate training from scratch (denoted as Scratch), or replacing LoRA modules with multiplayer perceptions (MLP) to demonstrate the efficiency of

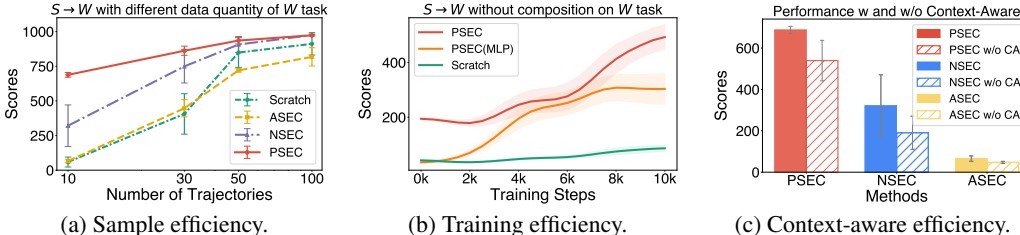

Figure 6: Comparisons on sample and training efficiency and the effectiveness of context-aware modular. S, W, R denote stand, walk and run, respectively. Each value is averaged over 10 episodes and 5 seeds.

compositions and LoRA module. *3)* We evaluate different PSEC variants without context-aware modular (denoted as w/o CA) to further highlight the crucial role of dynamically combining skills.

**Training and sample efficiency**. To demonstrate the training and sample efficiency of PSEC, we conduct extensive evaluations across varying numbers of trajectories and different methods. Figure 6(a) shows that PSEC achieves superior sample efficiency across different training sample sizes, particularly when data is scarce (e.g., only 10 trajectories). Figure 6(b) shows that PSEC can quickly attain excellent performance even without composition, highlighting the effectiveness of the LoRA modules. Hence, we train less than 50k gradient steps for almost all tasks, while previous methods typically require millions of gradient steps and data to obtain reasonable results.

**Continual Evolution**. Table 2 shows that PSEC effectively leverages prior knowledge to facilitate efficient policy learning given solely limited data. Notably, S+W→R outperforms S→R, demonstrating that the learning capability of PSEC gradually evolves as the skill library grows. In contrast, training from scratch or replacing the LoRA modules with MLP fails to learn new skills given limited data, highlighting the effectiveness of both utilizing prior knowledge and the introduction of LoRA to efficiently adapt to new skills and self-evolution. Moreover, note that even PSEC (MLP) outperforms NSEC and ASEC, further highlighting the advantages of parameter-level compositions.

Table 2: Results in policy shift setting. S, W, R denote stand, walk and run. 10 trajectories are provided for W and R tasks

|  | S→ W | S→ R | S+W→ R |
|---|---|---|---|
| Scratch | 58.9 | 25.5 | 25.5 |
| ASEC | 65.7 | 24.3 | 30.8 |
| NSEC | 320.9 | 38.5 | 39.4 |
| PSEC (MLP) | 424.1 | 143.3 | 194.5 |
| PSEC | **688** | **221** | **247** |

**Context-aware Composition *v.s.* Fixed Composition**. We carefully tune the fixed composition (w/o CA) of different skills during composition. However, Figure 6(c) shows that the context-aware modular can consistently outperform the fixed ones across different levels of compositions. This demonstrates the advantages of the context-aware composition network to fully leverage the prior knowledge in the skill library to enable efficient policy adaptations.

## 4.3 DYNAMICS SHIFT SETTING

We evaluate PSEC in another practical setting to further validate its versatility, where the dynamics $\mathcal{P}$ shift to encompass diverse scenarios such as cross-embodiment (O'Neill et al., 2024), sim-to-real transfer (Tobin et al., 2017), and policy learning in non-stationary environments (Xue et al., 2024).

**Setup**. We evaluate on the D4RL environments (Fu et al., 2020) , where we modify the dynamics and morphology of locomotive robots to reflect the dynamic changes. Specifically, we first pretrain $\pi_0$ using a dataset $\mathcal{D}_o^{\mathcal{P}_0}$ collected from a modified dynamics $\mathcal{P}_0$ and then equip it with a new small dataset $\mathcal{D}_o^{\mathcal{P}_1}$ collected under the original D4RL dynamics $\mathcal{P}_1$. *Friction*, *Thigh Size* and *Gravity* denote $\mathcal{P}_0$ modifies the friction condition, the thigh size of cheetah/walker, and the gravity respectively. Based on the new small dataset $\mathcal{D}_o^{\mathcal{P}_1}$, we set $w(s, a) = \exp(A_r^*(s, a))$ with $A_r^*(s, a)$ as the advantage function trained by expectile regression on $\mathcal{D}_o^{\mathcal{P}_1}$ (Kostrikov et al., 2022) to obtain a new policy $\pi_1$ and then optimize the context-aware composition network $\alpha(s; \theta)$ to combine $\pi_{0,1}$ to collaboratively work under dynamics $\mathcal{P}_1$. See Appendix C.3 for details.

**Baselines**. One branch of baselines consists in training $\pi_1$ from scratch on the small dataset $\mathcal{D}_o^{\mathcal{P}_1}$, which may face data scarcity challenges, including BC, offline RL methods like CQL (Kumar et al., 2020), IQL (Kostrikov et al., 2022), MOPO (Yu et al., 2020c). In addition, we evaluate some gen-

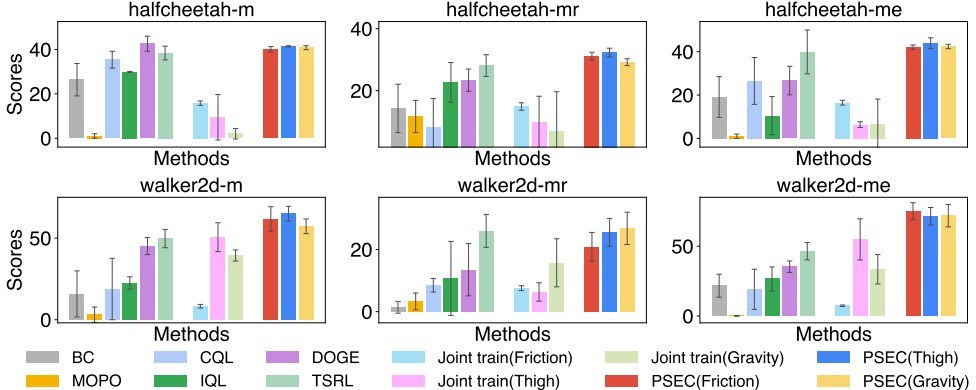

Figure 7: Results in the dynamics shift setting over 10 episodes and 5 seeds. -m, -mr and -me refer to $\mathcal{D}_o^{\mathcal{P}_1}$ sampling from medium, medium-replay and medium-expert-v2 data in D4RL (Fu et al., 2020), respectively.

eralizable offline RL methods including DOGE (Li et al., 2023) and TSRL (Cheng et al., 2023) that are superior in the small sample regimes. Additionally, we evaluate the policy trained on the combination of $\mathcal{D}_o^{\mathcal{P}_0}$ and $\mathcal{D}_o^{\mathcal{P}_1}$, referred to as Joint train, to show the advantages of the PSEC framework over a brute-force method of combining all data to address dynamic gaps.

**Main Results**. Figure 7 demonstrates that PSEC effectively utilizes transferable knowledge from the pretrained policy $\pi_0$ to enhance performance under changed dynamics. In contrast, traditional offline RL methods perform poorly with limited data in new dynamic settings. Moreover, PSEC surpasses specialized sample-efficient offline RL methods like TSRL and DOGE, showcasing its superior ability to leverage prior knowledge for increased training efficiency.

## 4.4 ABLATION STUDY

We primarily ablate on different LoRA ranks $n$ to assess the robustness of our methods in continual policy shift setting. Figure 8 demonstrates that under varied LoRA $n$ ranks, PSEC consistently outperforms the MLP variant across various LoRA ranks, demonstrating the superior robustness of LoRA modules. Among the different rank settings, we observe that $n = 8$ gives the best results and is therefore chosen as the default choice for the experiments. We hypothesize that using a rank greater than 8 degenerates because the training data is quite limited (e.g., only 10 demonstrations).

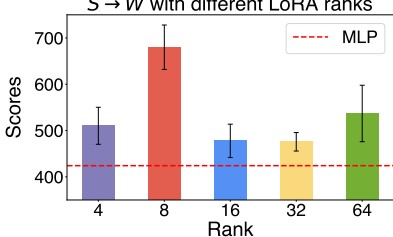

Figure 8: Ablations on LoRA ranks.

## 5 CONCLUSION

We propose PSEC, a framework that handles different skills as plug-and-play LoRA modules within an expandable skill library. This flexible approach enables the agents to reuse prior knowledge for efficient new skill acquisition and to progressively evolve in response to new challenges like humans. By exploiting the interpolation property of LoRA, we propose a context-aware compositional network that adaptively activates and blends different skills directly in the parameter space by merging the corresponding LoRA modules. This parameter-level composition enables the exploitation of more shared and complementary information across different skills, allowing for optimal compositions that collaboratively generate complex behaviors in dynamical environments. PSEC demonstrates exceptional effectiveness across diverse practical applications, such as multi-objective composition, continual policy shift and dynamic shift settings, making it highly versatile for real-world scenarios where knowledge reuse and monotonic policy improvements are crucial. One limitation is the pretrained policy $\pi_0$ may encompass diverse distributions to ensure good LoRA tuning. However, this can be mitigated by utilizing the broad out-of-domain dataset to enhance distribution coverage. More discussions on limitations and future works can be found in Appendix A.

## ACKNOWLEDGEMENT

This work is supported by National Key Research and Development Program of China under Grant (2022YFB2502904), National Natural Science Foundation of China under Grant 62403483, Grant U24A20279, and Grant U21A20518, and funding from Wuxi Research Institute of Applied Technologies, Tsinghua University under Grant 20242001120.

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

## A    LIMITATIONS AND FUTURE WORKS

In this section, we provide detailed discussions about the limitations and their potential solutions.

- **Assumption on the expressiveness of the pretrain policy**. The main limitation of PSEC is the assumption that the pre-trained $\pi_0$ covers a diverse distribution, which allows for efficient fine-tuning using small add-on LoRA modules. If this assumption does not hold, learning new skills through parameter-efficient fine-tuning may prove challenging, as significantly more parameters might be required to acquire new skills.

  *Potential solutions*: Note that this assumption is mild in relevant papers that utilize LoRA to learn new skills (Hu et al., 2021; Liu et al., 2023c). To tackle this problem, one straightforward solution is to increase the value of LoRA ranks to increase the learning capabilities of the newly introduced modules. Another simple solution is to leverage the cheap and abundant out-of-domain data to enhance the distribution coverage of the pretrained $\pi_0$ to enable efficient LoRA adaptations.

- **Redundant skill expansion**. In this paper, PSEC includes policies for all tasks in the skill library across its lifelong time. Although we adopt LoRA to reduce computational burden and memory usage, maintaining an extensive library of skill primitives may still lead to substantial computational costs.

  *Potential solutions*: Note that not all skills should be incorporated into the skill library, particularly those that are redundant and can be synthesized from other primitives. An interesting direction for future research is to develop an evaluation metric to assess the interconnections between different skills, such as the skill diversity (Pertsch et al., 2021; Eysenbach et al.), to only include essential, non-composable atomic primitives. Such a strategy could significantly reduce the management costs associated with maintaining the skill library.

- **Hyperparameter-tuning**: Another limitation is PSEC introduces another LoRA modules to learn new skills, which can introduce additional hyperparameters required to be tuned.

  *Potential solutions*: This limitation is widely existed in relevant works that try to reuse prior knowledge to learn new skills (Liu et al., 2023c; Clark et al., 2024; Wang et al., 2024d; Peng et al., 2019; Barreto et al., 2018), since almost all papers require additional parameters or regularization to adapt to the new skills. In this paper, we have ablated the robustness of PSEC against varied LoRA ranks, and demonstrate consistent superiority over the naive MLP modules in Figure 8, highlighting the robustness of PSEC for hyperparameter tuning.

- **Simple context-aware compositional modular**: We employ a simple context-aware modular $\alpha(s; \theta)$ to dynamically combine different primitives. This operation is simple and may not fully leverage the shared structure across skills for the target task.

  *Potential Solutions*: However, in our paper, we have demonstrated the superior advantages of this simple context-aware modular, as shown in Figure 6c. One interesting future direction is to adopt a more advanced model architecture, training objective, or more flexible gating approach to optimize the modular.

## B    DISCUSSIONS ON MORE RELATED WORKS

**Tabula Rasa**. Tabula rasa learning is one popular paradigm for diverse existing decision-making applications, such as robotics and games (Silver et al., 2017; Andrychowicz et al., 2020; Berner et al., 2019; Gong et al., 2024a; Vinyals et al., 2019). It directly learns policies from scratch without the assistance of any prior knowledge. However, it suffers from notable drawbacks related to poor sample efficiency and constraints on the complexity of skills an agent can acquire (Agarwal et al., 2022).

**Finetune-based Methods**. Some finetune-based methods aim to accelerate policy learning by leveraging prior knowledge. This knowledge may come from pretrained policy or offline data (Liu et al., 2024b; Wang et al., 2024a; Dai et al., 2024; Gong et al., 2024b; Cheng et al., 2024a), such as Offline-to-online RL (Nair et al., 2020; Lee et al., 2022b; Agarwal et al., 2022; Cheng et al., 2024b) and transfer RL (Barreto et al., 2018; Li et al., 2019; Lan et al., 2022). Some methods maintain a policy

library that contains pretrained policies and adaptively selects one policy from this set to assist policy training (Kim et al., 2024; Wang et al., 2024c; Barreto et al., 2018). However, they are generally restricted to single-task scenarios where all policies serve the same task (Zhang et al., 2023a), or only sequentially activate one policy in the pretrained sets, which greatly limits the expressiveness of the pretrained primitives (Li et al., 2019). Our method, on the contrary, can both leverage multi-task knowledge to fulfill the new task, and can simultaneously activate all skills to compose more complex behaviors.

**MoE in decision making**. The recent SDP (Wang et al., 2024d) is particularly relevant to our work. Specifically, SDP employs Mixture of Experts (MoE) (Shazeer et al., 2016) to encode skills as flexible combinations of forward path gated by distinct routers, allowing for efficient adaptation to new tasks by fine-tuning newly introduced expert tokens and task-specific routers. However, SDP necessitates that the pretrained policy $\pi_0$ be modeled with MoE layers, which imposes additional requirements on the model architecture. In contrast, our approach does not impose any constraints on the structure of the pretrained network and allows for the direct incorporation of new skills as plug-and-play LoRA modules. Moreover, when we identify a skill that is underperforming, we can easily modify the skill library by simply removing its plug-and-play LoRA modules. In contrast, using MoE limits this flexibility in managing different skills, making it challenging to mitigate the side effects caused by suboptimal skills. Therefore, PSEC offers a more flexible approach to managing the skill library, making it more feasible to scale up and incorporate a larger number of skills.

**LoRA in decision making**. Other relevant works such as TAIL (Liu et al., 2023c), LoRA-DT (Huang et al., 2024) and L2M (Schmied et al., 2024) also employ LoRA to encode skills. However, they solely investigate the parameter-isolation property of LoRA to prevent catastrophic forgetting, while overlooking the potential to merge different LoRA modules to interpolate new skills. Moreover, TAIL only studies the IL domain, L2M and LoRA-DT only study the RL domain, while PSEC both explore the effectiveness in RL and IL settings.

**LoRA for composition in other domains**. (Ponti et al., 2023; Clark et al., 2024; Huang et al., 2023; Zhong et al., 2024; Prabhakar et al., 2024) use LoRA for multi-task learning but using a fixed combination of LoRA modules, focusing on static settings like language model or image generation, thus limiting its expressiveness of the pretrained LoRA modules and flexibility of composition. In contrast, PSEC combines different LoRA via a context-aware modular, maximizing the expressiveness of pretrained skills to flexibly compose new skills, which is crucial for decision making since the real-time adjustment is required to handle the dynamical problems as shown in Figure 5.

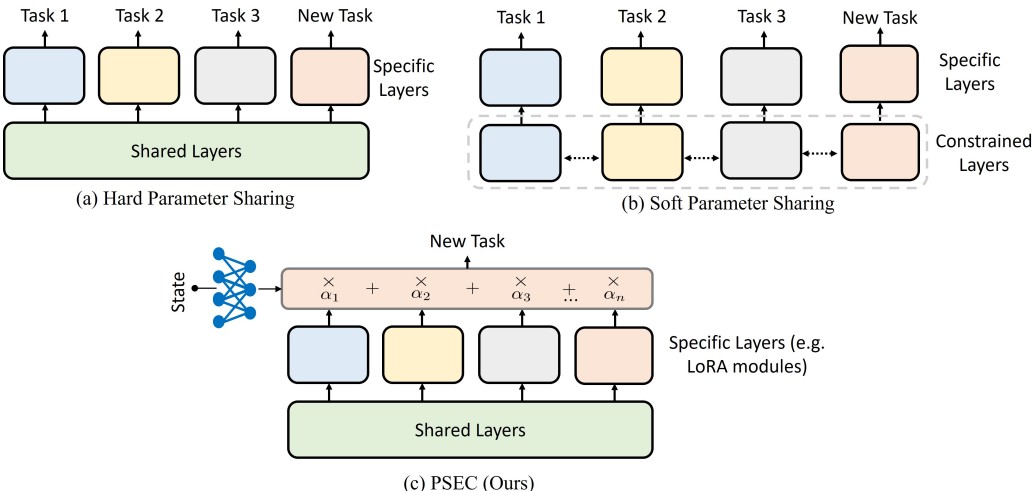

Figure 9: Illustrative comparisons between PSEC and other modularized multitask learning frameworks when deployed to continual learning settings.

**Modularized skills for multitask learning.** Multitask learning methods attempt to leverage the complementary benefits and commonalities across different tasks to enhance the cross-task generalization and capabilities (Wang et al., 2024d; Yang et al., 2020; Sun et al., 2022; 2023; Ruder, 2017;

Ma et al., 2024b;a). To achieve effective skill sharing, two primitive paradigms are introduced, including *Hard Parameter Sharing* and *Soft Parameter Sharing* (Ruder, 2017), as shown in Figure 9. All these methods demonstrate a *modularized structure*, where separate parameters are required to solve different tasks. Not only enjoying the benefits of multitask learning, this modularized design allows for efficient adaptation to new tasks by exploiting the shareable knowledge stored in different modules (Happel & Murre, 1994; SHARKEY, 1996; Auda & Kamel, 1998; 1999; Sodhani et al., 2022; Andreas et al., 2016; Alet et al., 2018; Ponti et al., 2023; Clark et al., 2024; Huang et al., 2023; Zhong et al., 2024; Prabhakar et al., 2024).

Hard parameter sharing approaches (Caruana, 1993; Sun et al., 2022; 2023; Baxter, 1997; León et al., 2021) aim to learn a shared feature that is strong and generalizable enough to capture the commonalities across all different tasks. This is achieved by developing a multi-head style structure, where different heads solve different tasks and all heads share some common layers (Sun et al., 2022; 2023; León et al., 2021; Bakker & Heskes, 2003). In this structure, zero-shot generalization to new tasks becomes possible if the shared layers can capture some generic features, following the spirits of meta learning (Finn et al., 2017; Gordon et al., 2019; Naik & Mammone, 1992; Lan et al., 2022). PSEC can be regarded as one specific type of hard parameter sharing method since different LoRA modules exploit a shared $\pi_0$. However, note that each LoRA module in PSEC is sequentially and independently optimized, thus making it easier to capture the task-specific features and avoid the potential gradient conflicts across different skills (Yu et al., 2020a; Liu et al., 2021). Previous methods, however, may introduce some gradient conflicts across different tasks that impede policy learning (Sun et al., 2022; Yang et al., 2020; Caruana, 1997), or suffer from collapsing to an entropic state and fail to encode task-specific features (Ponti et al., 2023).

Soft parameter sharing approaches (Yang et al., 2020; Wang et al., 2024d; Liu et al., 2024a; Duong et al., 2015; Yang & Hospedales, 2016; Ruder, 2017) are similar to the hard ones, with the differences primarily in the shared features. Instead of directly employing shared layers (Bakker & Heskes, 2003; Caruana, 1993; Sun et al., 2022; León et al., 2021), soft parameter sharing approaches adopt regularizations to enforce a "shared" feature across tasks, such as minimizing the L2 distance or cosine similarity across the features for different tasks (Duong et al., 2015; Yang & Hospedales, 2016; Ruder, 2017), adopting flexible structures like MoE layers (Shazeer et al., 2016; Yuksel et al., 2012; Wang et al., 2024d), soft modular (Yang et al., 2020), or resorting moving average across different features (Liu et al., 2024a; Lawson & Qureshi, 2024). These methods enjoy more flexibility than hard parameter sharing but may suffer from potential instability caused by improper regularizations and outlier tasks. For instance, Liu et al. (2024a); Lawson & Qureshi (2024) may undergo performance degradation without appropriate averaging weights if they are trying to combine a suboptimal skill learned on limited data.

**Modularized skills for continual learning and compositions.** More critically, the modularized design naturally facilitates continual evolvement by absorbing new skills in new modules in a parameter-isolation manner (Sodhani et al., 2022). This is one key advantage of modularized skills over traditional continual learning approaches since methods like EWC (Kirkpatrick et al., 2017), Rehearsal (Rolnick et al., 2019), Functional Regularization (Pan et al., 2020) often exhibit some catastrophic forgetting. The modularization method, however, can address this problem fundamentally by learning new parameters without disrupting pretrained ones. Along this line, numerous works also adopt modularized structure in a hard or soft manner as we discussed earlier (Ring, 1994; Pape et al., 2011; Huang et al., 2023; Andreas et al., 2016; Alet et al., 2018; Clark et al., 2024; Zhong et al., 2024; Prabhakar et al., 2024; Liu et al., 2024a) like PSEC. However, PSEC differs fundamentally in three key axes, including *how to obtain different modules*, *how to compose modules*, and *where to compose modules*.

- **How to obtain different modules?** Many previous methods typically assume a fixed set of modules during pretraining and jointly train all modules at once following a multitask learning paradigm (Ring, 1994; Pape et al., 2011; Schwarz et al., 2018; Ponti et al., 2023; Alet et al., 2018). Although this joint training approach enjoys the potential to exploit more shared features across tasks. The learned modules may fail to capture task-specific features, becoming general-purpose features and collapsing to highly entropic status, if the data distribution is very diverse and many outlier tasks exist (Ruder, 2017; Ponti et al., 2023). In contrast, PSEC independently trains each LoRA by exploiting a shared, frozen, and general-purpose $\pi_0$, avoiding lots of conflicts across different tasks and avoiding the

risks of collapsing (Yu et al., 2020a; Sun et al., 2022). We conduct empirical evaluations in our rebuttal to demonstrate this.

- **How to compose modules?** PSEC can iteratively expand its skill library to include more skills and then combine them to form complex ones, which is one common advantage of all modularized approaches. So, previous works can also iteratively expand their modules to encode new skills and then compose the pretrained ones to tackle new tasks, such as (Ring, 1994; Morrow & Khosla, 1997; Pape et al., 2011; Ponti et al., 2023; Alet et al., 2018; Huang et al., 2023; LeCun et al., 2006; Liu et al., 2022; Du et al., 2023). However, most previous works typically resort to a simple fixed combination of different modules, such as manually tuned weights (Liu et al., 2022; Du et al., 2023), thus significantly limiting the flexibility to handle decision-making scenarios where real-time composition adjustment is required to satisfy the dynamic demands. For instance, $2^n = C_n^0 + C_n^1 + C_n^2 + ... + C_n^n$ skills could be composed of n different (non-redundant) skills by using binary compositional weight (0 for deactivate and 1 for activate). So, naively adopting a fixed combination of different skills can be very suboptimal. In contrast, PSEC introduces a context-aware composition to dynamically combine different skills, greatly enhancing the expressiveness of the skill libraries by interpolating or extrapolating across different skills.

- **Where to compose modules?** Another key problem that should be investigated is where to compose different modules. Directly in the original output space (noise space (Ren et al., 2024; Zhang et al., 2023b) or action space (Peng et al., 2019; Qureshi et al., 2020; LeCun et al., 2006) or the parameter space (Huang et al., 2023; Prabhakar et al., 2024; Pape et al., 2011; Zhong et al., 2024). PSEC systematically investigates the advantages of skill compositions in parameter space over the noise space and action space, offering clear guidance for future research to expand and compose skills in parameter spaces rather than noise/action spaces. Also, intuitively, Figure 9 shows that PSEC holds the potential to exploit more complementary features or commonalities across tasks than naive hard parameter sharing or soft parameter sharing. Specifically, PSEC can fully leverage information across tasks to facilitate new task learning by employing the compositional network to combine all available parameters. Hard/Soft parameter sharing, however, must rely on a well-performed shared feature produced by the shared layers while discarding all other heads (Liu et al., 2024a; Lawson & Qureshi, 2024).

Some works use logical options for skill composition (Araki et al., 2021) but require significant human effort for skill management, limiting scalability. Additionally, Araki et al. (2021) focuses on efficient pretraining, not on fast adaptation/continual improvement. In contrast, PSEC targets the later setups and minimizes human effort by incorporating new skills as LoRA modules, which are then combined through auto-learned compositional networks.

# C  EXPERIMENTAL SETUPS

## C.1  MULTI-OBJECTIVE COMPOSITION

**Training details of PSEC**. In this setting, we have four networks required to train: the behavior policy $\pi_0$, the safety policy $\pi_1$ that minimizes the cost, the reward policy $\pi_2$ that maximizes the return, and the context-aware modular $\alpha(s; \theta) \in \mathbb{R}^2$. For each task, we first pretrain $\pi_0$ parameterized by $W_0$ as behavior policy by minimizing the following objective on the full DSRL dataset $\mathcal{D}$ (Liu et al., 2023a) to ensure a diverse pretrained distribution coverage:

$$\mathcal{L}_{\pi_0}(W_0) = \mathbb{E}_{t \sim \mathcal{U}, \epsilon \sim \mathcal{N}(0,I), (s,a) \sim \mathcal{D}} \left[ \left\| \epsilon - \epsilon_{W_0} \left( \sqrt{\bar{\rho}_t} a + \sqrt{1 - \bar{\rho}_t} \epsilon, t, s \right) \right\|^2 \right]. \quad (10)$$

Then, we equip the agent with the same dataset $\mathcal{D}$ but provide feasible label $h$ and reward labels $r$, forming the dataset $\mathcal{D}^h = \{(s, a, h, s')\}$ and $\mathcal{D}^r \{(s, a, r, s')\}$. Then we train $\pi_1$ and $\pi_2$ based on these datasets by optimizing their newly introduced LoRA modules $\Delta W_1$ and $\Delta W_2$ via minimizing the following objectives in Eq. (11-12):

$$\mathcal{L}_{\pi_1}(\Delta W_1) = \mathbb{E}_{t \sim \mathcal{U}, \epsilon \sim \mathcal{N}(0,I), (s,a) \sim \mathcal{D}^h} \left[ w^h(s, a) \left\| \epsilon - \epsilon_{W_1} \left( \sqrt{\bar{\rho}_t} a + \sqrt{1 - \bar{\rho}_t} \epsilon, t, s \right) \right\|^2 \right], \quad (11)$$

$$\mathcal{L}_{\pi_2}(\Delta W_2) = \mathbb{E}_{t \sim \mathcal{U}, \epsilon \sim \mathcal{N}(0,I),(s,a) \sim \mathcal{D}^r} \left[ w^r(s,a) \left\| \epsilon - \epsilon_{W_2} \left( \sqrt{\bar{\rho}_t} a + \sqrt{1 - \bar{\rho}_t} \epsilon, t, s \right) \right\|^2 \right], \quad (12)$$

where the weights of LoRA augmented layer are $W_1 = W_0 + 16\Delta W_1$ and $W_2 = W_0 + 16\Delta W_2$ as defined in Eq. (5). $w^h(s,a) := \exp(-A_h^*(s,a))$ and $w^r(s,a) := \exp(A_r^*(s,a))$ are the weighting function derived from the optimal feasible value function $A_h^*(s,a) = Q_h^*(s,a) - V_h^*(s)$ and reward value function $A_r^*(s,a) = Q_r^*(s,a) - V_r^*(s,a)$, optimized via expectile regression following (Kostrikov et al., 2022; Zheng et al., 2024), where $Q_h^*(s,a)$ and $V_h^*(s)$ can be obtained via minimizing Eq. (13-14), $Q_r^*(s,a)$ and $V_r^*(s)$ can be obtained via minimizing Eq. (15-16):

$$\mathcal{L}_{V_h} = \mathbb{E}_{(s,a) \sim \mathcal{D}^h} \left[ L_{\text{rev}}^\tau \left( Q_h(s,a) - V_h(s) \right) \right], \tag{13}$$

$$\mathcal{L}_{Q_h} = \mathbb{E}_{(s,a,s',h) \sim \mathcal{D}^h} \left[ \left( ((1-\gamma)h(s) + \gamma \max\{h(s), V_h(s')\}) - Q_h(s,a) \right)^2 \right], \tag{14}$$

$$\mathcal{L}_{V_r} = \mathbb{E}_{(s,a) \sim \mathcal{D}^r} \left[ L^\tau \left( Q_r(s,a) - V_r(s) \right) \right], \tag{15}$$

$$\mathcal{L}_{Q_r} = \mathbb{E}_{(s,a,s',r) \sim \mathcal{D}^r} \left[ \left( r + \gamma V_r(s') - Q_r(s,a) \right)^2 \right]. \tag{16}$$

where $L^\tau(u) = |\tau - \mathbb{I}(u < 0)| u^2$ and $L_{\text{rev}}^\tau(u) = |\tau - \mathbb{I}(u > 0)| u^2$ with $\tau \in (0.5, 1)$. By doing so, $\pi_1$ and $\pi_2$ become one safety policy that avoids unsafe outcomes and one reward policy that tries to maximize the cumulative returns, respectively.

Then, we train our context-aware modular network $\alpha(s; \theta)$ to combine $\pi_{0,1,2}$ to collaboratively tackle the safe offline RL problem. We filter the Top-30 trajectories with the highest rewards and costs below 5 from the dataset $\mathcal{D}$ to form a small near-expert dataset $\mathcal{D}^*$ that obtains a good balance among distributional shift, reward maximization and safety constraint. Then, we train $\alpha(s; \theta)$ by minimizing the following imitation learning loss based on the $\mathcal{D}^*$:

$$\mathcal{L}(\theta) = \mathbb{E}_{t \sim \mathcal{U}, \epsilon \sim \mathcal{N}(0,I),(s,a) \sim \mathcal{D}^*} \left[ \left\| \epsilon - \epsilon_W \left( \sqrt{\bar{\rho}_t} a + \sqrt{1 - \bar{\rho}_t} \epsilon, t, s \right) \right\|^2 \right], \tag{17}$$

where $W = W_0 + \sum_{i=1}^2 \alpha_i(s; \theta) \Delta W_i$ as defined in Eq. (7).

We train $\pi_0$ for 1M gradient steps with a batch size of 2048 to ensure a good performance of $\pi_0$. Then, we only train $\pi_1$ and $\pi_2$ for 50K gradient steps, for the efficiency of LoRA modules. For $\alpha(s; \theta)$, we only train it for 1K gradient steps since all decomposed policies including $\pi_{0,1,2}$ are ready to be composed, which can significantly reduce the computational burden leveraging these pretrained policies. Summarized hyperparameters can be found in Table 9.

**Baselines**. For FISOR (Zheng et al., 2024), CDT (Liu et al., 2023b), COptiDICE (Lee et al., 2022a), CPQ (Xu et al., 2022) and BC, we adopt the results from FISOR (Zheng et al., 2024). For NSEC and ASEC results, we only change the compositional stages, and meanwhile keep all other training details the same to ensure a fair comparison. Specifically, the context-aware modular for NSEC is trained via the following reparameterization method instead of the one in Eq. (12):

$$\epsilon_{\text{NSEC}} = \epsilon_0 + \sum_{i=1}^2 \alpha_i(s; \theta) \epsilon_i, \tag{18}$$

where $\epsilon_{0,1,2}$ is generated from networks with layers of $W_0$, $W_1 = W_0 + 16\Delta W_1$ and $W_2 = W_0 + 16\Delta W_2$, respectively. We can see that the composition in Eq. (18) between skills happens in the noise space, and thus we denote it as NSEC (noise skill expansion and composition).

For ASEC, we directly compose the generated actions of different policies:

$$a_{\text{ASEC}} = a_0 + \sum_{i=1}^2 \alpha_i(s; \theta) a_i \tag{19}$$

where $a_{0,1,2}$ are the actions generated from the denoising process in Eq. (2) using the predicted noise $\epsilon_{0,1,2}$ generated by networks with layers of $W_0$, $W_1 = W_0 + 16\Delta W_1$ and $W_2 = W_0 + 16\Delta W_2$, respectively. The composition happens in action space, and thus we denote it as ASEC (action skill expansion and composition).

## C.2 CONTINUAL POLICY SHIFT

To evaluate PSEC's ability to continually evolving its capabilities when tackling new challenges, we conduct experiments on DeepMind Control Suite (DMC) (Tassa et al., 2018), where a walker agent is progressively required to stand, walk, and run, as shown in Figure 11. We use three expert datasets including walker-stand $\mathcal{D}_e^{\mathcal{T}_0}$, walker-walk $\mathcal{D}_e^{\mathcal{T}_1}$, and walker-run $\mathcal{D}_e^{\mathcal{T}_2}$, released by Bai et al. (2024) for the policy learning. Specifically, $\mathcal{D}_e^{\mathcal{T}_0}$, $\mathcal{D}_e^{\mathcal{T}_1}$ and $\mathcal{D}_e^{\mathcal{T}_2}$ contains 1000, 10 and 10 trajectories, respectively. $\mathcal{D}_e^{\mathcal{T}_1}$ and $\mathcal{D}_e^{\mathcal{T}_2}$ contain only a handful of data because we aim to test if the agent can leverage the knowledge from the standing skill to efficiently adapt to new tasks. We first pretrain $\pi_0$ on the large $\mathcal{D}_e^{\mathcal{T}_0}$ to obtain the basic standing policy via minimizing the following behavior cloning loss:

$$\mathcal{L}_{\pi_0}(W_0) = \mathbb{E}_{t \sim \mathcal{U}, \epsilon \sim \mathcal{N}(0,I), (s,a) \sim \mathcal{D}_e^{\mathcal{T}_0}} \left[ \left\| \epsilon - \epsilon_{W_0} \left( \sqrt{\bar{\rho}_t} a + \sqrt{1 - \bar{\rho}_t} \epsilon, t, s \right) \right\|^2 \right]. \quad (20)$$

**Stand→ Walk (S→ W) task**. Then, we can integrate the walking skill $\pi_1$ into the skill library $\Pi$ by optimizing the following objective:

$$\mathcal{L}_{\pi_1}(\Delta W_1) = \mathbb{E}_{t \sim \mathcal{U}, \epsilon \sim \mathcal{N}(0,I), (s,a) \sim \mathcal{D}_e^{\mathcal{T}_1}} \left[ \left\| \epsilon - \epsilon_{W_1} \left( \sqrt{\bar{\rho}_t} a + \sqrt{1 - \bar{\rho}_t} \epsilon, t, s \right) \right\|^2 \right], \quad (21)$$

where $W_1 = W_0 + 16\Delta W_1$. Then, we train a context-aware modular $\alpha^{\text{walk}}(s; \theta_1) \in \mathbb{R}$ to combine $\pi_0$ and $\pi_1$ to jointly tackle the walking task:

$$\mathcal{L}(\theta_1) = \mathbb{E}_{t \sim \mathcal{U}, \epsilon \sim \mathcal{N}(0,I), (s,a) \sim \mathcal{D}_e^{\mathcal{T}_1}} \left[ \left\| \epsilon - \epsilon_{W_{\text{walk}}} \left( \sqrt{\bar{\rho}_t} a + \sqrt{1 - \bar{\rho}_t} \epsilon, t, s \right) \right\|^2 \right], \quad (22)$$

where $W_{\text{walk}} = W_0 + \alpha^{\text{walk}}(s; \theta_1)\Delta W_1$. In this setting, we hope the final policy parameterized by $W_{\text{walk}}$ can outperform the naive policy that is trained from scratch on the small data $\mathcal{D}_e^{\mathcal{T}_1}$ to demonstrate the significance of utilizing the prior knowledge in $\pi_0$ for efficient task adaptation.

**Stand→Run (S→R) task**. Here, the adaptation for the running policy $\pi_2$ is similar. We can replace $W_1$ in Eq. (21) with $W_2 = W_0 + 16\Delta W_2$ and $\mathcal{D}_e^{\mathcal{T}_1}$ as $\mathcal{D}_e^{\mathcal{T}_2}$ to train $\pi_2$ parameterized by $\Delta W_2$. Additionally, we replace $W_{\text{walk}}$ in Eq. (22) as $W_{\text{run}} = W_0 + \alpha^{\text{run}}(s; \theta_2)\Delta W_2$ and $\mathcal{D}_e^{\mathcal{T}_1}$ as $\mathcal{D}_e^{\mathcal{T}_2}$ to train $\alpha^{\text{run}}(s; \theta_2)$ to combine $\pi_0$ and $\pi_2$ to generate the running skill.

**Stand+Walk→Run (S+W→R) task**. After obtaining $\pi_0$, $\pi_1$, and $\pi_2$, the composition for the running skill becomes very simple. We can replace $W_{\text{walk}}$ in Eq. (22) as $W = W_0 + \sum_{i=1}^{2} \alpha_i(s; \theta)\Delta W_i$ to train $\alpha(s; \theta) \in \mathbb{R}^2$ to combine $\pi_{0,1,2}$ to generate the running skill. In this setup, we aim to prove that utilizing the library that contains $\pi_{0,1,2}$ (S+W→R) can outperform $\pi_{0,2}$ (S→R) to show the learning capability of PSEC can gradually grow after incorporating more skill primitives.

We train $\pi_0$ for 1M gradient steps with a batch size of 1024 to ensure a good performance of $\pi_0$. Then, we only train $\pi_1$ and $\pi_2$ for 10K gradient steps with 10 trajectories thanks to the efficiency of LoRA. For $\alpha^{\text{walk}}(s; \theta), \alpha^{\text{run}}(s; \theta), \alpha(s; \theta)$, we only train them for 1K gradient steps since the decomposed policies including $\pi_{0,1,2}$ in the skill library are ready to be composed, which can significantly reduce the computational burden leveraging these pretrained policies. The summarized hyperparameters can be found in Table 10.

**Baselines.** We compare PSEC with other composition methods NSEC and ASEC, the Scratch method, and the variant PSEC (MLP). NSEC and ASEC train the context-aware modular represented by Eq. (18) and Eq. (19), respectively. Scratch method means training a policy from scratch by IDQL (Hansen-Estruch et al., 2023), since we build our model based on the IDQL method. PSEC (MLP) replaces the LoRA matrices with the MLP network in PSEC.

**Experimental setups for Figure 6**. For Figure 6(a), we evaluate the sample efficiency of PSEC framework. Specifically, we evaluate on the **S→ W** task with different data quantities of the **W** dataset $\mathcal{D}_e^{\mathcal{T}_1}$, including 10, 30, 50, and 100 trajectories, trained with 10K, 30K, 50K, and 100K training steps, respectively. We compare PSEC with other baselines to demonstrate the sample efficiency of parameter-level composition over other composition methods.

For Figure 6(b), we visualize the training curves of PSEC, PSEC (MLP) and Scratch for the **S→W** task trained solely on Eq. (21) without the composition in Eq. (22) to demonstrate the efficiency of LoRA modules over the naive MLPs and the efficiency to leverage pretrain policies. In this setting, $\mathcal{D}_e^{\mathcal{T}_1}$ contains 10 trajectories and we train each method for 10K training steps.

For Figure 6(c), w/o CA represents the compositional weight $\alpha$ is tuned by humans, rather than auto-generated by our context-aware modular $\alpha_\theta$. We compare PSEC, NSEC, ASEC with their corresponding w/o CA variants to further demonstrate the importance of dynamical compositions.

We conduct similar experiments on the **S→R** task and the results are presented in Figure 12. Note that the running skill is more difficult. PSEC shows marked superiority on this challenging setting.

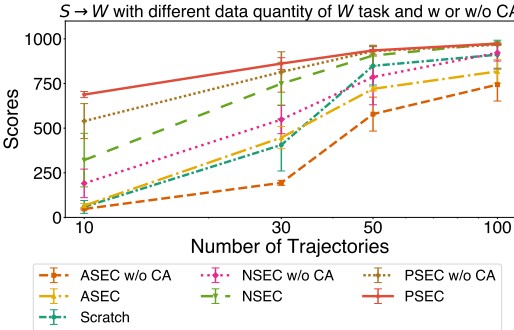

Figure 10: Results in the policy shift setting. Each value is averaged over 10 episodes and 5 seeds.

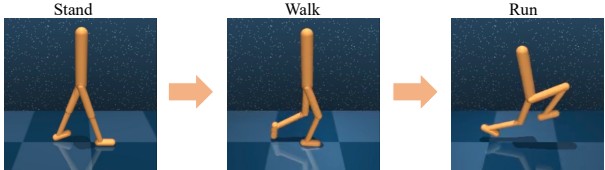

Figure 11: Continual evolution on DeepMind Control Suite for Continual policy shift.

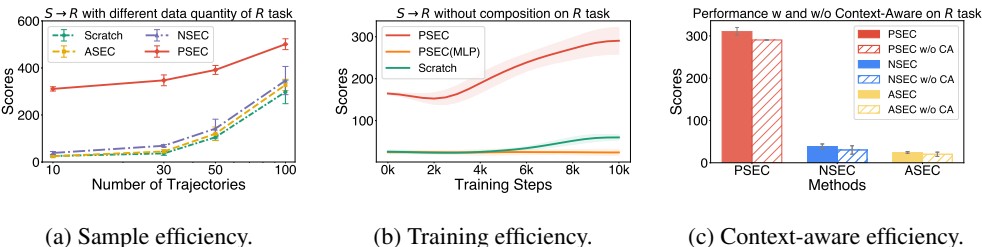

(a) Sample efficiency.      (b) Training efficiency.      (c) Context-aware efficiency.

Figure 12: Comparisons on sample and training efficiency and the effectiveness of context-aware modular. S, R denote stand, run, respectively. Each value is averaged over 10 episodes and 5 seeds.

## C.3 DYNAMIC SHIFT

To further validate the versatility of PSEC, we conduct experiments in a practical and common setting: dynamic shift. We conduct experiments on the D4RL benchmark, where we modify the dynamics and morphology of locomotive robots to reflect the dynamics changes. Our goal is to leverage the policies based on the source datasets $\mathcal{D}_o^{\mathcal{P}_0}$ and a small amount of the target datasets $\mathcal{D}_o^{\mathcal{P}_1}$ to adapt to the target task quickly. Specifically, the datasets $\mathcal{D}_o^{\mathcal{P}_0}$ contain 20K transitions with 3 types of dynamic modifications on $\mathcal{P}_0$: *1) Friction*: the friction coefficient of the robot is modified; *2) Gravity*: the gravity acceleration in the simulation environment is changed. *3) Thigh*: the thigh is enlarged to double its original size to produce a morphology gap on the embodiment. The target datasets $\mathcal{D}_o^{\mathcal{P}_1}$ are sampled from the D4RL benchmark with un-modified dynamics $\mathcal{P}_1$, including 6 types: halcheetah-medium-v2, halfcheetah-medium-replay-v2, halfcheetah-medium-expert-v2, walker2d-medium-v2, walker2d-medium-replay-v2, walker2d-medium-expert-v2, as shown in

Figure 13. Each dataset type of $\mathcal{D}_o^{\mathcal{P}_1}$ contains solely 10K transitions, which are too limited to train good policies directly on the target dynamics $\mathcal{P}_1$ from scratch.

We first pretrain $\pi_0$ with dataset $\mathcal{D}_o^{\mathcal{P}_0}$ for 20k training steps by behavior cloning via minimizing the following objectives:

$$\mathcal{L}_{\pi_0}(W_0) = \mathbb{E}_{t \sim \mathcal{U}, \epsilon \sim \mathcal{N}(0,I), (s,a) \sim \mathcal{D}_o^{\mathcal{P}_0}} \left[ \left\| \epsilon - \epsilon_{W_0} \left( \sqrt{\bar{\rho}_t} a + \sqrt{1 - \bar{\rho}_t} \epsilon, t, s \right) \right\|^2 \right]. \quad (23)$$

Then, we try to use the limited $\mathcal{P}_1$ to adapt $\pi_0$ to the target domain . PSEC uses LoRA to train a new policy $\pi_1$ with the pretrained source policy $\pi_0$ by minimizing the following objectives:

$$\mathcal{L}_{\pi_1}(\Delta W_1) = \mathbb{E}_{t \sim \mathcal{U}, \epsilon \sim \mathcal{N}(0,I), (s,a) \sim \mathcal{D}_o^{\mathcal{P}_1}} \left[ w^r(s,a) \left\| \epsilon - \epsilon_{W_1} \left( \sqrt{\bar{\rho}_t} a + \sqrt{1 - \bar{\rho}_t} \epsilon, t, s \right) \right\|^2 \right], \quad (24)$$

where $W_1 = W_0 + 16 \Delta W_1$. Finally, PSEC uses the context-aware modular $\alpha(s; \theta)$ to integrate policy $\pi_0, \pi_1$ using the target dataset $\mathcal{D}_o^{\mathcal{P}_1}$ to transfer to the target dynamics $\mathcal{P}_1$. The context-aware modular $\alpha(s; \theta)$ is trained for only 1k training steps by minimizing the following objectives:

$$\mathcal{L}(\theta) = \mathbb{E}_{t \sim \mathcal{U}, \epsilon \sim \mathcal{N}(0,I), (s,a) \sim \mathcal{D}_o^{\mathcal{P}_1}} \left[ \left\| \epsilon - \epsilon_W \left( \sqrt{\bar{\rho}_t} a + \sqrt{1 - \bar{\rho}_t} \epsilon, t, s \right) \right\|^2 \right], \quad (25)$$

where $W = W_0 + \alpha(s; \theta) \Delta W_1$ as defined in Eq. (7).

We train $\pi_0$ for 1M gradient steps with a batch size of 1024 to ensure a good performance of $\pi_0$. Then, we only train $\pi_1$ for 20k gradient steps, for the efficiency of LoRA modules. For $\alpha(s; \theta)$, we only train it for 1K gradients steps since all decomposed policies including $\pi_{0,1}$ are ready to be composed, which can efficiently adapt to the target domain leveraging the pretrained source policies. Summarized hyperparameters can be found in Table 11.

**Baselines.** We compare PSEC with other methods in dynamic shift settings, including behavioral cloning (BC), offline RL approaches like CQL (Kumar et al., 2020), IQL (Kostrikov et al., 2022), and model-based methods such as MOPO (Yu et al., 2020c). Additionally, we evaluate more generalizable offline RL methods, specifically DOGE (Li et al., 2023) and TSRL (Cheng et al., 2023), which have demonstrated superiority in small sample regimes. The baseline results for comparison are sourced from the TSRL paper (Cheng et al., 2023), which reports state-of-the-art performance in these regimes. Furthermore, we assess policies trained on combinations of the offline datasets $\mathcal{D}_o^{\mathcal{P}_0}$ and $\mathcal{D}_o^{\mathcal{P}_1}$ under various dynamics settings, referred to as Joint train (Gravity), Joint train (Friction), and Joint train (Thigh). These combinations involve training with one source dataset under dynamic shifts (e.g., changes in gravity, friction, or thigh size) and target datasets such as halfcheetah-medium-v2, halfcheetah-medium-replay-v2, halfcheetah-medium-expert-v2, walker2d-medium-v2, walker2d-medium-replay-v2, and walker2d-medium-expert-v2. In order to maintain fairness, the joint train method is trained in the same way as PSEC is trained on the source datasets. The results and training curves of PSEC across these settings are presented in Table 8 and Figure 20, respectively. These comparisons showcase the effectiveness of PSEC under dynamic shifts and small sample conditions.

## C.4 T-SNE EXPERIMENTAL SETUPS FOR FIGURE 4

To provide empirical support of the advantages of parameter-level composition over other levels of composition, we visualize the t-SNE (Van der Maaten & Hinton, 2008) projection of data samples in different spaces. Specifically, for each dataset $\mathcal{D}_e^{\mathcal{T}_0}, \mathcal{D}_e^{\mathcal{T}_1}, \mathcal{D}_e^{\mathcal{T}_2}$ in the continual policy shift setting in Section C.2, we randomly sample 512 data samples $(s, a)$, which forms three types of data that encode the standing, walking and running skill, respectively. In the action space, we directly utilize t-SNE projection to map these sampled data into a 2-dimensional space in Figure 4 (c). For the noise space, we add 1 step of noise on the sampled actions following the forward diffusion process in Eq. (1) and get the tuple $(s, a_1)$ for different skills. Then, we generate the noise based on this noisy tuples and visualize their t-SNE projections in Figure 4 (b). In parameter-space, we feed the noisy

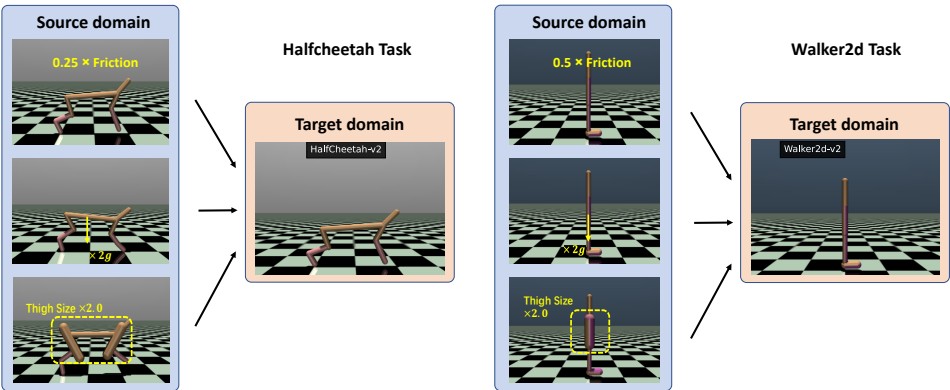

Figure 13: The illustration of the source and target domains for the dynamic shift setting.

tuples $(s, a_1)$ into the trained networks and get the output features of the middle LoRA augmented layers. Then, we project these features using t-SNE in Figure 4 (a).

# D MORE EXPERIMENTAL RESULTS

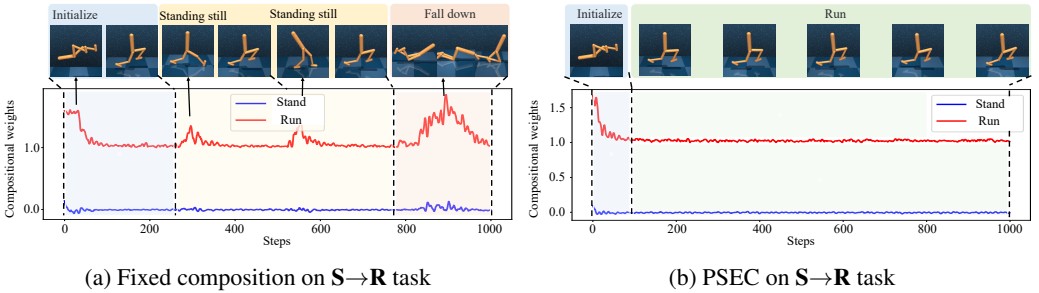

(a) Fixed composition on $\mathbf{S} \rightarrow \mathbf{R}$ task        (b) PSEC on $\mathbf{S} \rightarrow \mathbf{R}$ task

Figure 14: Outputs weights of the context-aware modular on DeepMind Control

## D.1 THE EFFECTIVENESS OF THE CONTEXT-AWARE MODULAR

**Context-aware modular for the continual policy shift.** To further explore the effectiveness of the context-aware module, we employ it to analyze the trajectories generated by policies composed using fixed compositional weights. Specifically, for the $\mathbf{S} \rightarrow \mathbf{R}$ task in Section C.2, the fixed composition method denote $W_{\text{run}} = W_0 + \alpha \Delta W_2$, which uses a fixed $\alpha = 16$ to compose $\pi_0$ and $\pi_2$. Figure 14 (a) shows that naively using fixed compositional weights might accidentally stuck in some local suboptimal behavior such as standing still or falling down. We can clearly observe that our context-aware modular provides corresponding responses to correct these undesired behaviors. Therefore, it is necessary to adjust the weights of different strategies to fit the current states. Figure 14 (b) presents the trajectories generated by PSEC. It clearly demonstrates that by utilizing the context-aware modular, the agent can make subtle adjustments between skills and stably run across the entire episodes.

## D.2 THE PARAMETER EFFICIENCY OF PSEC

**Parameter efficiency.** To evaluate the parameter efficiency of PSEC, we compare its parameter count and performance on various tasks against both the Scratch method and PSEC (MLP). The parameter count for PSEC includes the LoRA parameters and context-aware parameters specific to the walker-walk or walker-run tasks. The Scratch method represents training the policy from scratch with standard MLP. PSEC (MLP), which substitutes the LoRA weights with a standard MLP and retains the context-aware modular, has a higher parameter count than the Scratch method. The

parameter counts are illustrated in Figure 15. In terms of performance, the results from the Deep-Mind Control Suite (DMC) tasks, as shown in Figures 6 (b) and 12 (b), indicate that PSEC achieves significantly better performance despite having only 7.58% of the parameters used in the Scratch method. This performance advantage over both the Scratch method and PSEC (MLP) demonstrates that PSEC possesses strong parameter efficiency, effectively leveraging a smaller number of parameters for superior task performance. In this way, PSEC can leverage and expand upon its existing knowledge base in novel situations to enhance learning efficiency and adaptability.

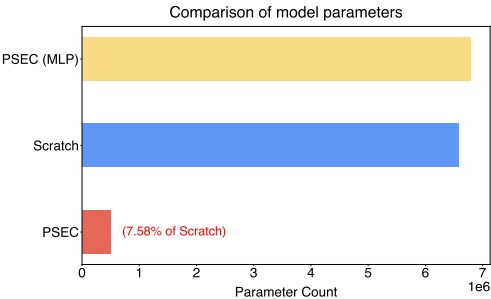

Figure 15: Comparison of Model Parameters: The parameter count for PSEC is approximately 7.58% of Scratch, demonstrating a significantly smaller model size while maintaining strong parameter efficiency, effectively leveraging a smaller number of parameters for superior task performance..

## E  MORE EXPERIMENTAL DETAILS

### E.1  DESCRIPTION OF TASKS

We conduct experiments on 9 MetaDrive tasks and 8 Bullet-Safety-Gym tasks in the DSRL benchmark (Liu et al., 2023a). The visualization of the environments is shown in Figure 16. The tasks aim to learn policy from different level datasets such that the policy satisfies a safety constraint (normalized cost < 1) and achieves higher rewards.

**MetaDrive.** It leverages the Panda3D game engine to simulate realistic driving scenarios. The tasks are categorized as {Road}{Vehicle}, where "Road" encompasses three levels of difficulty for self-driving cars: easy, medium, and hard, while "Vehicle" represents four levels of surrounding traffic density: sparse, mean, and dense. In MetaDrive's autonomous driving tasks, costs are incurred from three safety-critical scenarios: (i) collision, (ii) out of road, and (iii) over-speed.

**Bullet-Safety-Gym.** The environments are built on the PyBullet physics simulator. They feature four types of agents: Ball, Car, Drone, and Ant, alongside two task types: Circle and Run. Tasks are designated as {Agent}{Task}, combining the agent and the corresponding task type.

### E.2  ILLUSTRATION OF THE RECORDED DATA

To get a more intuitive look at the recorded data, we calculate the total reward and total cost for each trajectory in the datasets. These values are then plotted on a two-dimensional plane, where the x-axis corresponds to the total cost and the y-axis to the total reward. The results are shown in Figure 18 in the Appendix E of the paper. The plot highlights the dataset's diversity, particularly in how it captures a range of trajectory behaviors. The reward frontiers relative to cost illuminate the task's complexity, as the shape of these frontiers can significantly influence the challenges faced by offline learners. Trajectories offering high rewards but incurring high costs pose an alluring yet risky opportunity, often testing the balance between optimizing performance and maintaining safety constraints. This duality underscores the importance of robust algorithms that can navigate the trade-off effectively.

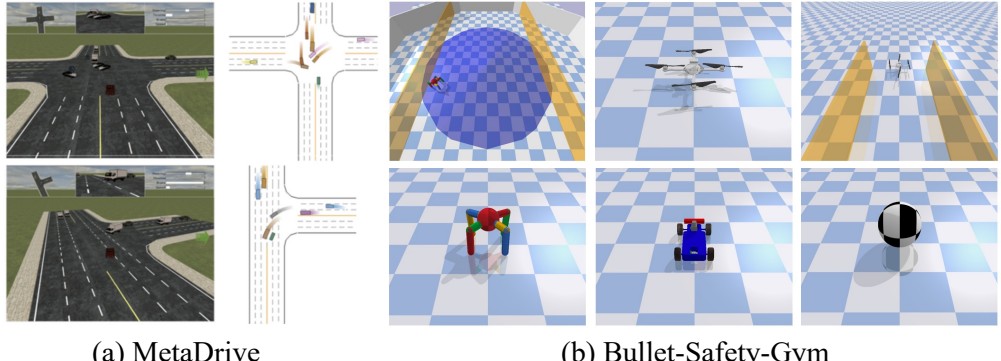

(a) MetaDrive        (b) Bullet-Safety-Gym

Figure 16: Visualization of the simulation environments and representative tasks of MetaDrive and Bullet-Safety-Gym. The figure is credited to Liu et al. (2023a).

### E.3 ADVANTAGE OF THE BENCHMARK

By generating diverse datasets across many environments with systematically varied complexities, the DSRL benchmark creates a rich and representative evaluation suite. This diversity ensures that our method is tested under a wide range of conditions, capturing different task structures, safety constraints, and levels of stochasticity. Meanwhile, the DSRL benchmark includes multiple objectives, making it well-suited for testing the flexibility and efficiency of our method in handling new tasks. Providing diverse datasets across varying difficulty levels and incorporating multiple optimization goals enables a comprehensive evaluation of our method's adaptability and performance across a broad spectrum of scenarios.

## F MORE EXPERIMENTS ON META-WORLD

To evaluate the effectiveness of PSEC on more complex experiments, we conduct experiments on Meta-World benchmark (Yu et al., 2020b), which consists of 50 diverse tasks for robotic manipulation, such as grasping, manipulating objects, opening/closing a window, pushing buttons, locking/unlocking a door, and throwing a basketball. We compare PSEC with the strong baseline L2M (Schmied et al., 2024). Next, we will elaborate on the three experiment settings in our paper.

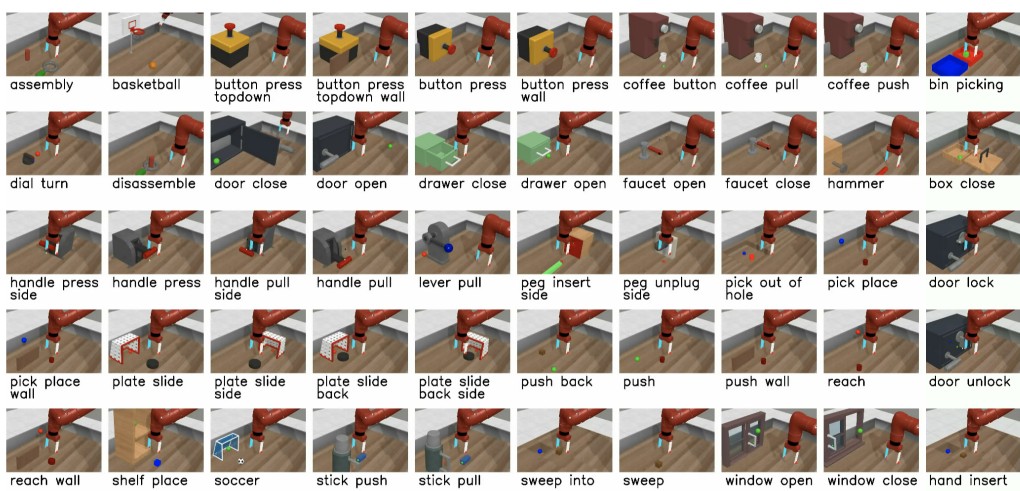

Figure 17: Visualization of the simulation environments and representative tasks of Meta-World.

## F.1 CONTINUAL LEARNING SETTING

Following Continual world (Wolczyk et al., 2021) and L2M (Schmied et al., 2024), we split the 50 tasks into 40 pre-training tasks and 10 fine-tuning unseen tasks (CW10). The training datasets are the same as the datasets collected by L2M. We train 10K steps per task in CW10, which is only 10% training steps of L2M, with a batch size of 1024. After every 10K update steps, we switch to the next task in the sequence. Then we evaluate it on all tasks in the task sequence. The results are shown in Table 3 and Table 4. We compare the performance of PSEC with L2M and other strong baselines. Thanks to the efficiency of skill composition in parameter space, PSEC can substantially outperform all L2M variants in a large margin, demonstrating that PSEC can achieve better performance on complex tasks.

Table 3: Success rates of different methods.

| Methods | Success Rate |
|---|---|
| L2M | 0.65 |
| L2M-oracle | 0.77 |
| L2P-Pv2 | 0.40 |
| L2P-PreT | 0.34 |
| L2P-PT | 0.23 |
| EWC | 0.17 |
| L2 | 0.10 |
| PSEC (Ours) | **0.87** |

Table 4: Performance of PSEC on different tasks.

| Tasks | PSEC |
|---|---|
| peg-unplug-side-v2 | 0.87 |
| window-close-v2 | 0.88 |
| shelf-place-v2 | 0.85 |
| push-v2 | 0.89 |
| handle-press-side-v2 | 0.95 |
| stick-pull-v2 | 0.74 |
| push-back-v2 | 0.85 |
| faucet-close-v2 | 0.92 |
| push-wall-v2 | 0.86 |
| hammer-v2 | 0.91 |
| **Mean** | **0.87** |

## F.2 UNSEEN TASKS SETTING

To further evaluate the efficiency of PSEC on more challenging tasks, we pretrain on fewer (18) tasks and evaluate it on more (12) unseen tasks than the first setting. Firstly, we pretrain and finetune 18 tasks to obtain 18 LoRA modules. The performance on the 18 pretrained tasks is reported in Table 5. We compare the performance of PSEC with Scratch, ASEC and NSEC methods. The results show that PSEC can achieve enhanced skill learning even when the pretrained model is combined with one LoRA for each task if the skill is composed in parameter space. Then, we evaluate PSEC with the obtained 18 LoRA modules on the unseen tasks. For the unseen tasks, we conduct two types of experiments: few-shot setting and zero-shot setting.

**Few-shot.** We perform few-shot learning by training the context-aware modular for 1k steps using only 10% of the total available data for unseen tasks. This setup simulates scenarios with limited data on new tasks. The results, summarized in Table 6, demonstrate that PSEC achieves a high success rate on unseen tasks. This indicates that PSEC can effectively adapt to new tasks, showcasing its capability for rapid transfer learning and efficient adaptation in data-scarce environments.

**Zero-shot.** No data from the unseen tasks is used to train the context-aware modular. Instead, the modular is trained for 2k steps using datasets from 18 pre-trained tasks. It is then evaluated directly on 12 unseen tasks, utilizing 4 seeds and 10 episodes per task. The results are shown in Table 7. Interestingly, even without access to unseen task data during training, PSEC demonstrates strong performance on several tasks. Notably, PSEC substantially outperforms NSEC and ASEC on this zero-shot transfer setting, highlighting the advantages of skill compositions in parameter spaces over noise and action spaces. Overall, the results demonstrate PSEC's ability to effectively utilize knowledge from previously learned skills to achieve strong zero-shot transfer.

Table 5: Performance comparison on 18 pretrained tasks.

| Tasks | Scratch | ASEC | NSEC | PSEC |
|---|---|---|---|---|
| peg-insert-side-v2 | 0.50 | 0.87 | 0.88 | **0.90** |
| peg-unplug-side-v2 | 0.35 | 0.61 | 0.78 | **0.86** |
| button-press-topdown-v2 | 0.71 | 0.88 | 0.88 | **0.89** |
| push-back-v2 | 0.26 | 0.61 | 0.76 | **0.88** |
| window-close-v2 | 0.65 | 0.84 | 0.84 | **0.88** |
| door-open-v2 | 0.74 | 0.85 | **0.86** | **0.86** |
| handle-press-v2 | 0.67 | 0.96 | **0.97** | **0.97** |
| plate-slide-side-v2 | 0.27 | 0.23 | 0.53 | **0.74** |
| handle-pull-side-v2 | 0.76 | 0.94 | 0.94 | **0.95** |
| window-open-v2 | 0.87 | 0.75 | 0.88 | **0.89** |
| door-close-v2 | 0.90 | 0.89 | 0.89 | **0.91** |
| reach-v2 | 0.89 | **0.95** | **0.95** | **0.95** |
| push-v2 | 0.15 | 0.58 | 0.81 | **0.92** |
| stick-push-v2 | 0.44 | 0.54 | 0.17 | **0.79** |
| drawer-close-v2 | **0.97** | **0.97** | **0.97** | **0.97** |
| plate-slide-back-v2 | 0.90 | 0.94 | 0.94 | **0.95** |
| coffee-button-v2 | 0.91 | 0.94 | 0.94 | **0.95** |
| hand-insert-v2 | 0.30 | 0.68 | 0.63 | **0.89** |
| **Mean** | 0.62 | 0.78 | 0.81 | **0.90** |

Table 6: Few-shot performance comparison on 12 unseen tasks.

| Tasks | ASEC | NSEC | PSEC |
|---|---|---|---|
| plate-slide-v2 | 0.14 | 0.66 | **0.89** |
| handle-press-side-v2 | 0.73 | 0.65 | **0.92** |
| button-press-wall-v2 | 0.09 | 0.03 | **0.72** |
| button-press-topdown-wall-v2 | 0.87 | 0.88 | **0.89** |
| push-wall-v2 | 0.57 | 0.68 | **0.88** |
| reach-wall-v2 | 0.41 | 0.36 | **0.90** |
| faucet-close-v2 | 0.41 | 0.49 | **0.90** |
| button-press-v2 | 0.02 | 0.14 | **0.23** |
| plate-slide-back-side-v2 | 0.17 | 0.19 | **0.92** |
| handle-pull-v2 | 0.15 | 0.21 | **0.93** |
| faucet-open-v2 | 0.14 | 0.16 | **0.89** |
| stick-pull-v2 | 0.00 | 0.00 | **0.32** |

# G  MORE VISUALIZATION OF ADVANTAGES OF PSEC OVER NSEC AND ASEC

To test whether the newly learned skills effectively utilize the shared knowledge of previous skills, we evaluate the running policy obtained through context-aware modular combined with standing and walking skills on three rewards: stand, walk, and run. If the running skill can still get a relatively high stand or walk reward, this represents the final combined running skill retaining these previous skills. We compare PSEC with other composition methods ASEC and NSEC. For each method, we rollout 10K steps and record the three rewards. The summarized rewards can be found in Figure 19. The results show that PSEC achieves high rewards across all tasks, whereas NSEC and ASEC cannot, demonstrating that the PSEC's running skill retains behaviors from walking and standing and suggesting superior skill sharing of PSEC compared to NSEC and ASEC.

Table 7: Zero-shot performance comparison on 12 unseen tasks.

| Tasks | ASEC | NSEC | PSEC |
|---|---|---|---|
| plate-slide-v2 | 0.03 | 0.00 | **0.15** |
| handle-press-side-v2 | 0.50 | 0.60 | **0.62** |
| button-press-wall-v2 | 0.00 | 0.00 | **0.40** |
| button-press-topdown-wall-v2 | 0.85 | 0.87 | **0.89** |
| push-wall-v2 | 0.53 | 0.53 | **0.71** |
| reach-wall-v2 | 0.34 | 0.05 | **0.90** |
| faucet-close-v2 | 0.00 | 0.00 | **0.16** |
| button-press-v2 | 0.00 | 0.00 | **0.15** |
| plate-slide-back-side-v2 | 0.00 | 0.00 | 0.00 |
| handle-pull-v2 | 0.00 | 0.00 | 0.00 |
| faucet-open-v2 | 0.00 | 0.00 | **0.77** |
| stick-pull-v2 | 0.00 | 0.00 | 0.00 |

Table 8: Results in the dynamics shift setting over 10 episodes and 5 seeds. -m, -mr and -me refer to $\mathcal{D}_o^{\mathcal{P}_1}$ sampling from medium, medium-replay and medium-expert V2 data in D4RL (Fu et al., 2020), respectively.

| Metric | Halfcheetah-m | Halfcheetah-mr | Halfcheetah-me | Walker2d-m | Walker2d-mr | Walker2d-me |
|---|---|---|---|---|---|---|
| BC | $26.4 \pm 7.3$ | $14.3 \pm 7.8$ | $19.1 \pm 9.4$ | $15.8 \pm 14.1$ | $1.4 \pm 1.9$ | $21.7 \pm 8.2$ |
| MOPO | $-1.1 \pm 4.1$ | $11.7 \pm 5.2$ | $-1.1 \pm 1.4$ | $3.1 \pm 4.7$ | $3.3 \pm 2.7$ | $0.1 \pm 0.3$ |
| CQL | $35.4 \pm 3.8$ | $8.1 \pm 9.4$ | $26.5 \pm 10.8$ | $18.8 \pm 18.8$ | $8.5 \pm 2.19$ | $19.1 \pm 14.4$ |
| IQL | $29.9 \pm 0.2$ | $22.7 \pm 6.4$ | $10.5 \pm 8.8$ | $22.5 \pm 3.8$ | $10.7 \pm 11.9$ | $26.5 \pm 8.6$ |
| DOGE | $\mathbf{42.6 \pm 3.4}$ | $23.4 \pm 3.6$ | $26.7 \pm 6.6$ | $45.1 \pm 10.2$ | $13.5 \pm 8.4$ | $35.3 \pm 4.1$ |
| TSRL | $38.4 \pm 3.1$ | $28.1 \pm 3.5$ | $39.9 \pm 21.1$ | $49.7 \pm 10.6$ | $26.0 \pm 11.3$ | $46.4 \pm 13.2$ |
| Joint train(Gravity) | $2.0 \pm 1.4$ | $6.8 \pm 3.9$ | $6.8 \pm 5.4$ | $39.4 \pm 3.4$ | $15.7 \pm 7.7$ | $33.5 \pm 10.5$ |
| Joint train(Friction) | $15.8 \pm 1.0$ | $14.9 \pm 1.2$ | $16.5 \pm 1.1$ | $8.3 \pm 1.1$ | $7.6 \pm 0.8$ | $7.4 \pm 0.5$ |
| Joint train(Thigh) | $9.5 \pm 5.3$ | $9.8 \pm 8.5$ | $6.4 \pm 1.3$ | $50.6 \pm 8.8$ | $6.3 \pm 3.0$ | $54.9 \pm 14.8$ |
| **Dynamic shift** | | | | | | |
| PSEC(Gravity) | $40.8 \pm 0.9$ | $29.2 \pm 1.1$ | $42.4 \pm 1.0$ | $57.2 \pm 4.5$ | $\mathbf{26.8 \pm 5.2}$ | $71.8 \pm 8.0$ |
| PSEC(Friction) | $40.1 \pm 1.2$ | $31.1 \pm 1.3$ | $42.1 \pm 1.0$ | $61.7 \pm 7.5$ | $20.9 \pm 4.6$ | $\mathbf{75.0 \pm 12.1}$ |
| **Body shift** | | | | | | |
| PSEC(Thigh) | $\mathbf{41.4 \pm 0.3}$ | $\mathbf{32.3 \pm 1.4}$ | $\mathbf{43.9 \pm 2.5}$ | $\mathbf{64.96 \pm 4.5}$ | $25.5 \pm 4.5$ | $71.4 \pm 14.3$ |

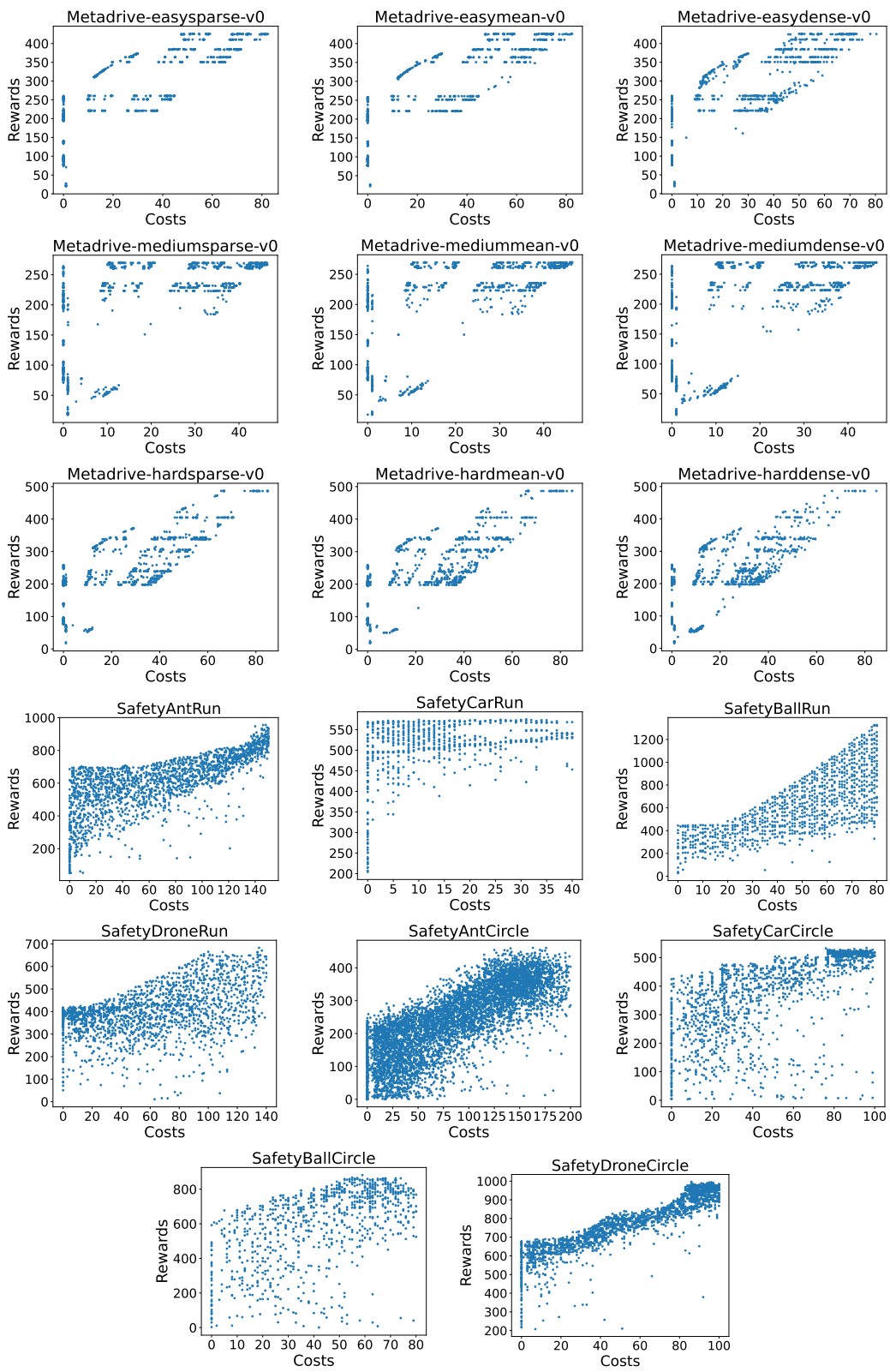

Figure 18: Illustration of the cost-reward plot for datasets from MetaDrive and Bullet-Safety-Gym.

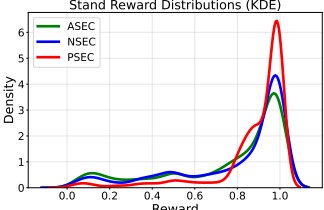 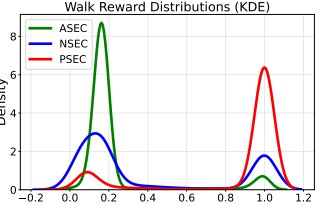 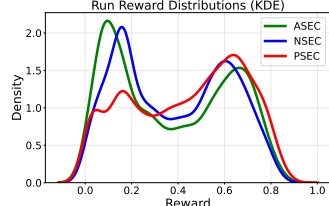

Figure 19: We evaluate the final running policies of PSEC, NSEC and ASEC with the "stand," "walk," and "run" rewards with 10 episodes and 3 random seeds. Then we plot the reward distribution by kernel density estimation (KDE). Each curve represents the probability density of rewards obtained for a specific reward. The results show that PSEC achieves high rewards across all tasks, whereas NSEC and ASEC cannot, demonstrating that the PSEC's running skill retains behaviors from walking and standing and suggesting superior skill sharing of PSEC compared to NSEC and ASEC.

Table 9: Hyperparameters for multi-objective composition tasks.

|  | Hyper-parameters | Value |
|---|---|---|
| shared hyperparameters | Normalized state | True |
| | Target update rate | 1e-3 |
| | Expectile $\tau$ | 0.9 |
| | Discount $\gamma$ | 0.99 |
| | Actor learning rate | 3e-4 |
| | Critic learning rate | 3e-4 |
| | Number of added Gaussian noise $T$ | 5 |
| $\pi_0$ | hidden dim | 256 |
| | hidden layers | 2 |
| | activation function | ReLU |
| | Mini-batch size | 2048 |
| | Optimizer | Adam (Kingma & Ba, 2014) |
| | Training steps | 1e6 |
| $\pi_1$ | $Q_r^*(s,a)$ hidden dim | 256 |
| | $Q_r^*(s,a)$ hidden layers | 2 |
| | $Q_r^*(s,a)$ activation function | ReLU |
| | $V_r^*(s)$ hidden dim | 256 |
| | $V_r^*(s)$ hidden layers | 2 |
| | $V_r^*(s)$ activation function | ReLU |
| | Actor hidden dim | 256 |
| | Actor hidden layers | 2 |
| | Actor Activation function | ReLU |
| | Mini-batch size | 2048 |
| | Optimizer | Adam |
| | Training steps | 5e4 |
| $\pi_2$ | $Q_h^*(s,a)$ hidden dim | 256 |
| | $Q_h^*(s,a)$ hidden layers | 2 |
| | $Q_h^*(s,a)$ activation function | ReLU |
| | $V_h^*(s)$ hidden dim | 256 |
| | $V_h^*(s)$ hidden layers | 2 |
| | $V_h^*(s)$ Activation function | ReLU |
| | Actor hidden dim | 256 |
| | Actor hidden layers | 2 |
| | Actor Activation function | ReLU |
| | Mini-batch size | 2048 |
| | Optimizer | Adam |
| | Training steps | 5e4 |
| $\alpha(s;\theta)$ | hidden dim | 256 |
| | hidden layers | 2 |
| | activation function | ReLU |
| | Mini-batch size | 2048 |
| | Optimizer | Adam |
| | Training steps | 1e3 |
| LoRA | rank $n$ | 8, 16 |

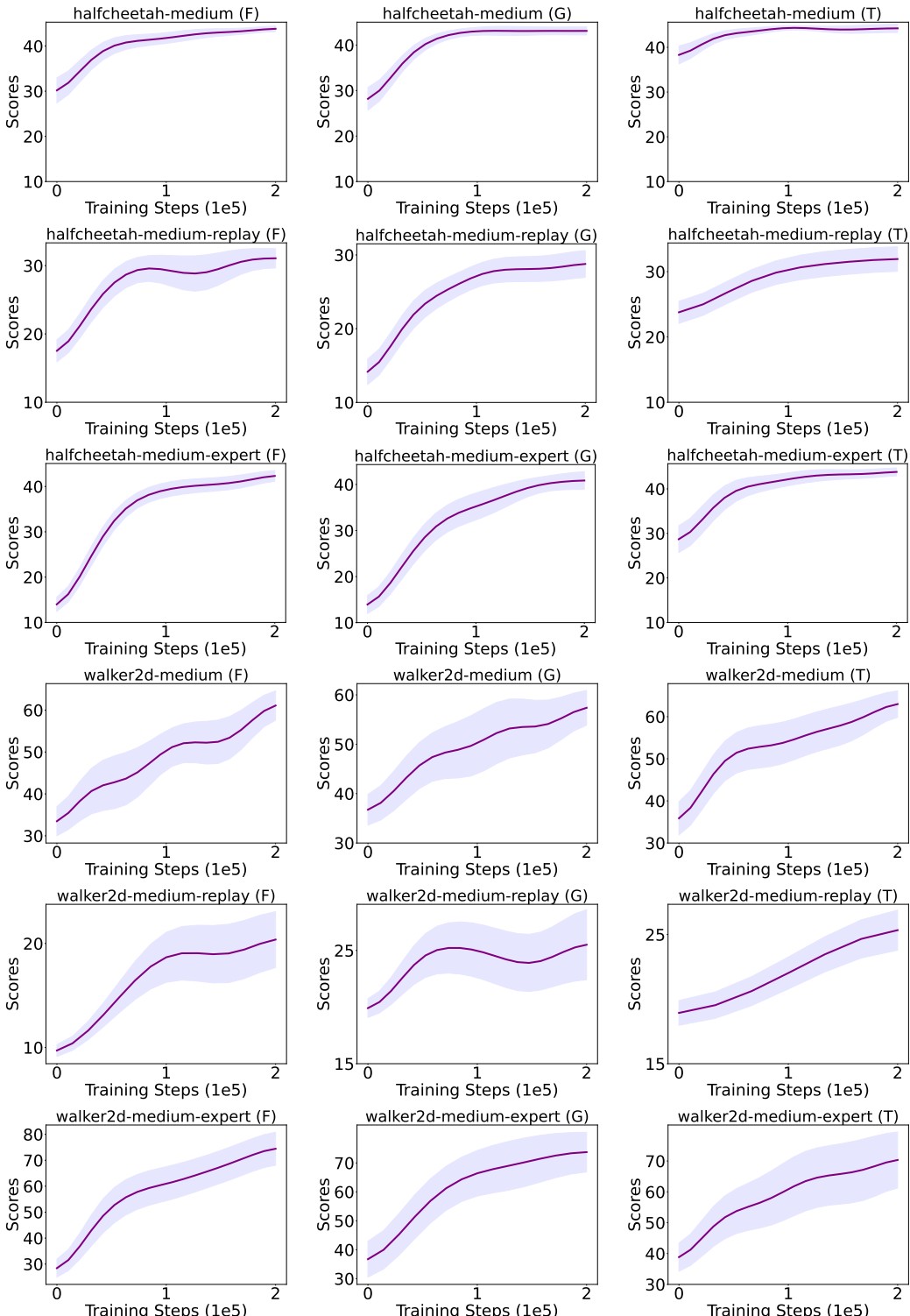

Figure 20: Results of performance conducted on dynamic shift and body shift tasks. The lines and shaded areas indicate the averages and standard deviations calculated over 5 random seeds.

Table 10: Hyperparameters for continual policy shift.

|  | Hyper-parameters | Value |
|---|---|---|
| shared hyperparameters | Normalized state | True |
|  | Target update rate | 1e-3 |
|  | Expectile $\tau$ | 0.9 |
|  | Discount $\gamma$ | 0.99 |
|  | Actor learning rate | 3e-4 |
|  | Critic learning rate | 3e-4 |
|  | Number of added Gaussian noise $T$ | 5 |
| $\pi_0$ | hidden dim | 256 |
|  | hidden layers | 2 |
|  | activation function | ReLU |
|  | Mini-batch size | 1024 |
|  | Optimizer | Adam |
|  | Training steps | 1e6 |
| $\pi_1$ | hidden dim | 256 |
|  | hidden layers | 2 |
|  | Activation function | ReLU |
|  | Mini-batch size | 1024 |
|  | Optimizer | Adam |
|  | Training steps | 1e4 |
| $\pi_2$ | Actor hidden dim | 256 |
|  | Actor hidden layers | 2 |
|  | Actor Activation function | ReLU |
|  | Mini-batch size | 1024 |
|  | Optimizer | Adam |
|  | Training steps | 1e4 |
| $\alpha(s; \theta)$ | hidden dim | 256 |
|  | hidden layers | 2 |
|  | activation function | ReLU |
|  | Mini-batch size | 1024 |
|  | Optimizer | Adam |
|  | Training steps | 1e3 |
| LoRA | rank $n$ | 8 |

Table 11: Hyperparameters for dynamic shift.

| | Hyper-parameters | Value |
|---|---|---|
| shared hyperparameters | Normalized state | True |
| | Target update rate | 1e-3 |
| | Expectile $\tau$ | 0.9 |
| | Discount $\gamma$ | 0.99 |
| | Actor learning rate | 3e-4 |
| | Critic learning rate | 3e-4 |
| | Number of added Gaussian noise $T$ | 5 |
| $\pi_0$ | hidden dim | 256 |
| | hidden layers | 2 |
| | activation function | ReLU |
| | Mini-batch size | 1024 |
| | Optimizer | Adam |
| | Training steps | 1e6 |
| $\pi_1$ | $Q_r^*(s,a)$ hidden dim | 256 |
| | $Q_r^*(s,a)$ hidden layers | 2 |
| | $Q_r^*(s,a)$ activation function | ReLU |
| | $V_r^*(s)$ hidden dim | 256 |
| | $V_r^*(s)$ hidden layers | 2 |
| | $V_r^*(s)$ activation function | ReLU |
| | Actor hidden dim | 256 |
| | Actor hidden layers | 2 |
| | Actor Activation function | ReLU |
| | Mini-batch size | 1024 |
| | Optimizer | Adam |
| | Training steps | 2e4 |
| $\alpha(s;\theta)$ | hidden dim | 256 |
| | hidden layers | 2 |
| | activation function | ReLU |
| | Mini-batch size | 1024 |
| | Optimizer | Adam |
| | Training steps | 1e3 |
| LoRA | rank $n$ | 8 |

