# OpenReview forum: "Skill Expansion and Composition in Parameter Space"
_ICLR.cc/2025/Conference — ICLR 2025 Poster_

### Official Review · Reviewer_oGS2 · 2024-10-31

**Soundness:** 3
**Presentation:** 4
**Contribution:** 3
**Rating:** 6
**Confidence:** 3

**Summary:**

This paper proposes Parametric Skill Expansion and Composition framework, which encodes skill primitives offline using LoRA modules and learning skill compositions with the generated LoRA library. The PSPEC framework is designed to cope with multi-objective composition as well as adapting to shift in policy and dynamics. The authors provides extensive experimentation on their proposed algorithm on the D4RL, DSRL, and DeepMind Control Suite.

**Strengths:**

- PSEC has a very intuitive and stright-forward design that utilizes LoRA structure. While PSEC wouldn't be the only structure to use the LoRA as building blocks for skills composition, the authors provides tSNE analysis to show why their method encodes skills better than the alternatives
- The paper is very pleasant to read and the paper structure too, is very straightforward. The authors also covers the related works very well which makes PSEC's design choices to be convincing.
- The experimentations are very extensive.
- I found the graphics very easy to understand and they summarize the key points very well. From intro to methods, I could very easily follow the keypoints by reading the details and captions in the graphics beforehand.

**Weaknesses:**

- LoRA is one of the key elements of the PSEC's structure, and while LoRA is mentioned in the abstract, the authors do not explain what LoRA is until page 2. I think it would be better to say in the abstract what LoRA stands for or simply refer to the module as a skills module if not explaining what LoRA is.
- While the experiments are extensive, I think some of the assumptions and motivations were not well explained. I have included the details in the questions section.
- I think one of the possible limitations of composing skills as a weighted some of LoRA actions is that some skills are not simply some of sub-skills. For example, the skill of throwing a ball is more of a fluid movements where the a person builds up the momentum and transfer the kinetic energy to the ball, instead of some combinations of walking forward and rotating torso. I am curious how the PSEC would perform with more complex tasks.

**Questions:**

- I think the paper lacks general motivations and overview on the benchmarks used. Why does this paper uses an offline safe RL benchmark instead of just an offline RL benchmarks in general? What are the tasks and what do the recorded data look in the offline benchmarks? Why would they provide a characteristic examples to highlight the pros and cons of PSEC?

---

> ### Author Response · Authors · 2024-11-20
> **Response to Reviewer oGS2**
>
> We sincerely appreciate the reviewer for the constructive comments on our work! Regarding the concerns from the reviewer oGS2, we provide the responses as follows:
>
> >**W1: More explanation about LoRA in Abstract...**
>
> - Thanks for this valuable comment! We have extended the discussion about LoRA in the Abstract. Please see the Abstract for details.
>
>
> >**W2&Q1.1: I think the paper lacks general motivations and overview on the benchmarks used...**
>
> Sorry for the confusion. Here, we summarize our motivations again. Specifically, we evaluate different axes of PSEC abilities through different benchmarks.
>
> - First, we evaluate whether the context-aware modular framework of PSEC can effectively compose multiple skills. We test this in a multi-object composition setting (Section 4.1) using a safe offline RL benchmark, where objectives like maximizing reward, minimizing cost, and staying close to behavior policy must be jointly addressed. Unlike traditional offline RL benchmarks that focus on a single objective, this setup better assesses PSEC's compositional capabilities.
> - Second, we assess whether PSEC can iteratively evolve as it incorporates more skills (Section 4.2). This is demonstrated using the DMC control suite, progressing from standing to walking and eventually running, showcasing PSEC's ability to continually learn and adapt.
> - Finally, we test PSEC's applicability across diverse scenarios (Section 4.3). To do this, we evaluate its performance in handling dynamics shifts by modifying the transition dynamics of an offline RL benchmark.
>
> Please refer to the first paragraph of each section (Section 4.1~4.3) for more details.
>
>
> >**W3. I think one of the possible limitations of composing skills as a weighted some of LoRA actions is that some skills are not simply some of sub-skills.**
>
> - Thanks for this valuable comment! The reviewer mentions an interesting skill of throwing a ball. This skill may not be composable from others but can still act as a sub-skill for more complex tasks, such as playing basketball. PSEC can address these non-composable skills by integrating them as new LoRA modules, allowing them to be combined for more complex tasks.
> - We provide more experimental results on more complex tasks in the Meta-World benchmark, covering a total of 50 skills. Please see `general response` and `Appendix F in our revised paper` for details.
>
>
>
> >**Q1.2&Q1.3: What are the tasks and what do the recorded data look in the offline benchmarks? Why would they provide a characteristic examples to highlight the pros and cons of PSEC?**
>
> - For the high-level motivations, please refer to the responses to `W2 & Q1` for details.
> - For benchmark details, we have added more illustrations and discussions about the safe offline RL in `Appendix E in our revised paper`. More details about the DMC control suite, dynamics shift, and Meta-World evaluations can be found in `Appendix C.2, C.3, F`, respectively. Please check them for details.
>
> We sincerely thank the reviewer for the valuable feedback. Please let us know if we have adequately addressed your concerns. If so, we would greatly appreciate it if the reviewer could consider adjusting the score.

---

> > ### Comment · Reviewer_oGS2 · 2024-11-25
> >
> > Thank you for your responses! I feel that my questions has been addressed and changed my score accordingly.

---

> ### Author Response · Authors · 2024-11-25
> **Thanks for increasing the score to a 6!**
>
> Thanks for the replay and for increasing the score to a 6! It's happy to hear that we've successfully addressed your concerns!

---

### Official Review · Reviewer_1bSW · 2024-11-01

**Soundness:** 4
**Presentation:** 4
**Contribution:** 3
**Rating:** 8
**Confidence:** 4

**Summary:**

This paper introduces a novel framework called Parametric Skill Expansion and Composition (PSEC). PSEC employs Low-Rank Adaptation (LoRA) to learn and store new skills, and directly synthesizes skills in the parameter space. It incorporates a context-aware module to adaptively compose skills in dynamic environments and utilizes diffusion models for policy modeling. The framework has been evaluated across various scenarios, including multi-objective composition, policy shifts, and dynamics changes.

**Strengths:**

The key features of PSEC include efficient skill learning and storage utilizing LoRA, direct skill synthesis in parameter space, adaptability through a context-aware module, and the ability for continuous skill expansion. Its effectiveness has been validated through various experiments, demonstrating versatility across different scenarios.

**Weaknesses:**

* While the strong assumption about the expressiveness of the pre-trained policy and the scalability issue of the skill library appear to be weaknesses, these seem to have been addressed in the paper's appendix.
* A potential limitation of the authors' proposed framework is its applicability only to environments where data is available. This could be considered a weakness of the paper. It is conceivable that if the framework could be extended to unseen tasks through transfer learning or fast adaptation techniques, it would demonstrate greater differentiation from previous research.
* Minor:
    * Generally, skill learning, as in  [1] and [2], refers to 'temporal abstraction', but the skill defined by the authors seems to differ from this. This may lead to confusion with existing skill learning concepts.

    * [1] Pertsch, Karl, Youngwoon Lee, and Joseph Lim. "Accelerating reinforcement learning with learned skill priors." Conference on robot learning. PMLR, 2021.
    * [2] Eysenbach, Benjamin, et al. "Diversity is all you need: Learning skills without a reward function." arXiv preprint arXiv:1802.06070 (2018).

**Questions:**

* One of the main strengths claimed for PSEC is that it "reduced computational costs and memory usage". How does this compare to the baselines?

---

> ### Author Response · Authors · 2024-11-20
> **Response to Reviewer 1bSW**
>
> We sincerely appreciate the reviewer for the constructive comments on our work! Regarding the concerns from the reviewer 1bSW, we provide the responses as follows:
>
> >**W2: A potential limitation of the authors' proposed framework is its applicability only to environments where data is available...**
>
> Thanks for this valuable comment!
>
> - We extend PSEC to a zero-shot generalization setting on the MetaWorld benchmark. Please see `general response` and `Appendix F in our revised paper` for details. Briefly, we jointly train PSEC's compositional network on 18 tasks and test PSEC on 12 unseen tasks. Results show that the compositional network can generalize well on unseen tasks, primarily due to the flexible combination and invocation of previously learned skills in the parameter space, which allows it to generalize to a certain extent to unseen scenarios.
> - In addition, all experiments in our paper inherently demonstrate few-shot adaptation/transfer, since the adaptations use solely limited data, such as the transfer from standing to walking to running, use only 10 trajectories, respectively.
> - To further illustrate this efficiency, we conduct a series of experiments on MetaWorld with more complex tasks. Please see `general response` and `Appendix F in our revised paper` for details.
>
> >**W3: ...this may lead to confusion with existing skill learning concepts.**
>
> - Good point! In this paper, each skill corresponds to a LoRA module for each task, unlike other works where skills are defined by temporal abstractions. PSEC’s skill definition allows for more flexible skill library expansion and management, since we can directly add new LoRA modules for new tasks. However, as discussed in `Appendix A (Redundant Skill Expansion)`, this can lead to redundant growth. To mitigate this, criteria for expansion can be implemented, such as assessing skill diversity, which is exactly mentioned by reviewer 1bSW in these relevant papers.
>
> >**Q1: How does PSEC compare to the baselines in terms of computational costs and memory usage?**
>
> - In the following table, we report the training time, steps and trainable parameters of PSEC and the SOTA baseline in the safe offline RL benchmark, FISOR, for illustration.
>
> | Metric          | FISOR     | PSEC    |
> | --------------- | --------- | ------- |
> | training time   | 1h 15m 39 | 13m 45  |
> | training step   | 1 M       | 0.101 M |
> | trainable parameter count | 9.7M   | 0.8 M  |
>
> - FISOR requires 1M training steps to train a stack of large MLPResNet blocks with 9.7M parameters. In contrast, PSEC only trains 3 small networks: 2 LoRA modules for reward maximization and cost minimization, and 1 compositional network, totaling just 0.8M parameters. This results in significantly fewer training steps and reduced training time for convergence.
> - Also, we've included some of these discussions in our first submission, please see `Appendix D.2` for details.
>
> We sincerely thank the reviewer for the valuable feedback. Please let us know if we have adequately addressed your concerns, and feel free to share any additional questions or comments.

---

> > ### Comment · Reviewer_1bSW · 2024-11-25
> > **Official Comment by Reviewer 1bSW**
> >
> > I appreciate the detailed responses from the authors. All my concerns have been addressed. I will keep my score as it is.

---

> > > ### Author Response · Authors · 2024-11-25
> > > **Thanks for the reply!**
> > >
> > > Thanks for the replay! It's happy to hear that we've successfully addressed your concerns!

---

### Official Review · Reviewer_18jx · 2024-11-03

**Soundness:** 2
**Presentation:** 2
**Contribution:** 3
**Rating:** 6
**Confidence:** 3

**Summary:**

This is a paper presents a new skill composition method, which
composes skills in the parameter space. Concretely, the paper
introduces PSEC, which fine-tunes new skills using low-rank adaptation
(LoRA) and argues the actions synthesized through this parameter-space
composition greatly improves the performance, compared to two other
composition methods termed noise-level composition and action-level
composition.

**Strengths:**

o This paper presents an interesting idea of how to compose skills in
the parameter space, and provide a clear categorization of skill
composition: parameter-level, noise-level, and action level. This
paper shows a way where neural network-based skills can be composed at
the parameter space level, instead of composing at the action output
level.

o In the long run, the proposed method can open up the possibility for
efficient skill learning based on large models in decision-making
domains

**Weaknesses:**

o While the general motivation in the first paragraph makes sense, the
example needs to be justified a bit better From online statistics, the
time difference between child learning to walk and learning to stand
without support is on average 2-2.5 months. Not everyone would call
this "rapid" (Ln 32)

o While the idea of parameter-level composition is interesting and
also introduces fewer bottlenecks compared to methods like action
level, the argument of parameter-level composition being superior is
not well-grounded. The explanation from line 256 to line 304 lacks
convincing evidence.

o The t-SNE plot (Figure 4) does not directly reveal that the
parameter space "shares knowledge" - if 4(a) can be explained to have
shared knowledge, why is 4(b) not a plot that shows "shared
knowledge"? The authors might try to design a more specific
visualization that clearly illustrates the idea of "shared
knowledge."

o In the experiment sections, there are a lot of terms that authors
either didn't specify clearly or misuse the words.

    o It is unclear what "versatility" means. Authors should provide a
    clear definition of this term(Line 354&473)

    o Line 397, the authors shouldn't use the word "generated" to
    describe other comparison methods, especially when the results
    haven't been revealed.

    o Line 374 "popular safe offline RL benchmark" is not
    informative. The authors should replace this with  an explanation
    of what problems does it contain (robot manipulation? autonomous
    driving? a collecition of them?) and why is it a good benchmark to
    test on (large-scale? diverse? or any other key features that make
    them special)

    o What does "safety" refer to? It's not explained well. The only
    thing I can take a guess is the cost that shows in the table,
    which does not really provide much information on how to interpret
    those numbers.

o The paper touches on multi-task learning and continual learning, and
uses a skill library.
All of these topics have been studied to some degree for decades.  Yet
the majority of citations are from 2023 and 2024, with only two
citations to work prior to 2016 - and those are just to energy-based
models and t-SNE.
Without putting the work in the proper context of prior work, its
novelty and significance can't be properly assessed.

o Minor: There are many small grammatical errors throughout the paper
- including in the first sentence of the abstract!

**Questions:**

o See the concerns and questions from Weaknesses

o If I understand correctly from the paragraph "Context-aware
Composition" (Line 244-254), only $W_{k}$ is updated at skill k. If
that's the case, What would be the impact of existing pre-trained LoRA
weights? Is it possible that because there are two skills that are
contradictory to each other, the LoRA fine-tuning on a new skill might
be harder to train? It will be good to see such experiments, and see
if the method still applies; or an additional experiment can conducted
where the learning curriculum is shuffled. Will the result be
consistent with the existing ones where learning happens from standing
to walking, running?

o Can this method scale to dozens of skills instead of three skills
presented in this paper? An experiment on stress testing the number of
skills it can handle will be able to support the effectiveness of the
proposed method.

---

> ### Author Response · Authors · 2024-11-20
> **Response to Reviewer 18jx (1/2)**
>
> We sincerely appreciate the reviewer for the constructive comments on our work! Regarding the concerns from the reviewer 18jx, we provide the responses as follows:
>
> >**W1. ...Not everyone would call this "rapid"**
>
> - Thanks for this valuable comment. We've revised our paper accordingly.
>
> >**W2. the argument of parameter-level composition being superior is not well-grounded ...**
>
> - We have provided qualitative experiments in Figure 4 and discussed them in line346-line351. The discussion from line 256 to line 304 is an intuitive explanation.
>
> >**W3: The t-SNE plot (Figure 4) does not directly reveal that the parameter space "shares knowledge"...**
>
> - Thanks for the valuable comment! We have added a new visualization in `Figure 19 in Appendix G of our revised paper`. This figure inspects whether PSEC’s final combined running skill retains behaviors from its component skills including standing and walking. Specifically, we summarize the run/walk/stand reward distributions for the trajectories rollouted by PSEC, NSEC, and ASEC. The results show that PSEC achieves high rewards across all tasks, demonstrating that its running skill retains behaviors from walking and standing, suggesting superior skill sharing compared to NSEC and ASEC.
> - Additionally, there seems to be some misunderstanding for the reviewer. Figure 4(b) illustrates that different skills are clustered together chaotically, indicating suboptimal grouping due to overly overlapping skills. In contrast, Figure 4(a) preserves the distinctions between different skills, showing disjoint spaces that maintain clear separations while also revealing a shared structure, suggesting potential shared knowledge across these skills.
>
> >**W4: a lot of terms that authors either didn't specify clearly ...**
>
> Thanks for these comments. We've revised them accordingly.
> - `versatility`. This highlights PSEC’s applicability across diverse settings, including multi-objective composition, continual RL, and dynamics shift scenarios.
> - `generated`. We have made the revisions; please refer to the updated paper for details.
> - `popular safe offline RL benchmark`. We selected a safe offline RL benchmark to test PSEC's ability to compose skills into complex ones. This benchmark requires jointly maximizing reward, minimizing cost, and avoiding distributional shift to solve the final problem—ideal for evaluating compositionality, as improper composition can lead to suboptimal results. Specifically, we evaluated 9 autonomous driving and 8 locomotion tasks from the DSRL benchmark, where each skill has clear physical meanings, as shown in Figure 5. See `Appendix E in the revised paper` for more details.
> - `Safety`. We clarified this in the caption of Table 1, following the DSRL benchmark’s definition of safety as a normalized cost below 1.
> - `related works`. We appreciate this comment and have expanded the discussion on related works in Appendix B in our revised paper. Specifically, to the best of our knowledge, our paper is the first that systematically investigates the advantages of skill composition and expansion in `parameter space` over noise and action space over 4 benchmarks, covering , offering a clear guidance for future reseaches in this field.
> - `grammatical errors`. We have carefully revised the paper to address grammatical issues and will continue to proofread it to minimize such errors.

---

> > ### Author Response · Authors · 2024-11-20
> > **Response to Reviewer 18jx (2/2)**
> >
> > >**Q1: If I understand correctly from the paragraph "Context-aware Composition" (Line 244-254), only is updated at skill k...**
> >
> > - Actually, the contradictory experiments mentioned by the reviewer were evaluated using the safe offline RL benchmark in Section 4.1, where the objectives of "maximizing reward, minimizing cost, and avoiding distributional shift" often conflict. For instance, the compositional network must prioritize safety over reward when approaching a potential collision. Results in Figure 5 show that our approach effectively handles this challenge.
> > - Also, there seems to be some misunderstanding. The context-aware compositional network is only updated after the new skill's LoRA module is trained. Specifically, PSEC involves three types of models: the pretrained model, pretrained LoRA modules for previous skills and a new LoRA module for current task. The new LoRA is fine-tuned upon the pretrained model only, not with existing LoRA modules. Once fine-tuning is complete, the context-aware compositional network integrates the pretrained model, existing LoRA modules and the new LoRA. This prevents conflicts between parameters of old and new LoRAs during training. Therefore, shuffling the curriculum will not change the performance of final performance of running.
> >
> > >**Q2: Can this method scale to dozens of skills...**
> > - Thanks for this valuable comments. We've added experiments on more complex skills on the Metaworld benchmark, covering a total of 50 skills. The results still demonstrate the effectiveness of PSEC. Please see `general response` and `Appendix F in our revised paper` for details.
> >
> > We sincerely thank the reviewer for the valuable feedback. Please let us know if we have adequately addressed your concerns. If so, we would greatly appreciate it if the reviewer could consider adjusting the score.

---

> ### Comment · Reviewer_18jx · 2024-11-24
>
> Thank you for your responses.  I'm inclined to raise my score a bit.  But for the paper to be considered for acceptance, I think it's essential that there be a more thorough treatment of related work, as indicated in my review.

---

> ### Author Response · Authors · 2024-11-25
> **Thanks for the replay!**
>
> We really appreciate your time engaged in the review and rebuttal phase and we feel our work has been greatly improved through your constructive comments! Regarding the remaining concerns on the literature review, we thoroughly examined over 30 modularized multitask learning and continual learning methods and tried our best to summarize the differences across three key axes: `how to obtain different modules`, `how to compose modules`, `where to compose modules`. Please see the revised `Appendix B (blud text)` for details. Due to time limits, we'll continue to include more relevant works and refine our writing to provide a more concise version.
>
> Here, we list some newly added references.
>
> [1] Meta-neural networks that learn by learning. 1992.
>
> [2] Design and evolution of modular neural network architectures. 1994.
>
> [3] Manipulation task primitives for composing robot skills. 1997.
>
> [4] A Bayesian/information theoretic model of learning to learn via multiple task sampling. 1997.
>
> [5] Multitask learning. 1997.
>
> [6] Modular neural network classifiers: A comparative study. 1998.
>
> [7] Modular neural networks: A survey. 1999.
>
> [8] Multitask learning: A knowledge-based source of inductive bias. 1993.
>
> [9] Combining artificial neural nets. 1996.
>
> [10] Task clustering and gating for Bayesian multitask learning. 2003.
>
> [11] Modular deep belief networks that do not forget. 2011.
>
> [12] Twenty years of mixture of experts. 2012.
>
> [13] Trace norm regularised deep multi-task learning. 2016.
>
> [14] Neural module networks. 2016.
>
> [15] Meta-learning for fast adaptation of deep networks. 2017.
>
> [16] Overcoming catastrophic forgetting in neural networks. 2017.
>
> [17] Modular meta-learning in abstract graph networks for combinatorial generalization. 2018.
>
> [18] Progress & compress: A scalable framework for continual learning. 2018.
>
> [19] Meta-learning probabilistic inference for prediction. 2018.
>
> [20] Experience replay for continual learning. 2019.
>
> [21] Gradient surgery for multi-task learning. 2020.
>
> [22] Continual deep learning by functional regularisation of memorable past. 2020.
>
> [23] Conflict-averse gradient descent for multi-task learning. 2021.
>
> [24] The logical options framework. 2021.
>
> [25] In a nutshell, the human asked for this: Latent goals for following temporal specifications. 2021.
>
> [26] An introduction to lifelong supervised learning. 2022.
>
> [27] Combining parameter-efficient modules for task-level generalisation. 2023.
>
> [28] Lorahub: Efficient cross-task generalization via dynamic lora composition. 2023.
>
> [29] Solving Continual Offline Reinforcement Learning with Decision Transformer. 2024.
>
> [30] Merging decision transformers: Weight averaging for forming multi-task policies. 2024.
>
>
> Thanks for your time and valuable feedback again! Please feel free to let us know if the reviewer has further comments. It would be really appreciated if the reviewer could consider re-evaluating our paper.

---

> > ### Comment · Reviewer_18jx · 2024-11-28
> >
> > Thank you.  I have raised my score.

---

> > > ### Author Response · Authors · 2024-11-28
> > > **Thanks for increasing the score!**
> > >
> > > Thanks for your feedback and for raising your score!

---

### Official Review · Reviewer_ac7Q · 2024-11-03

**Soundness:** 3
**Presentation:** 3
**Contribution:** 2
**Rating:** 6
**Confidence:** 4

**Summary:**

The paper introduces the Parametric Skill Expansion and Composition (PSEC) framework, which allows RL agents to learn and combine new skills by efficiently using a "skill library". Instead of relearning tasks from scratch, PSEC uses LoRA modules—compact, adaptable components—that can be added to this library as plug-and-play skills. This design enables agents to adapt to new tasks by combining skills directly within the model's parameters, allowing them to leverage shared knowledge from previous tasks while avoiding "catastrophic forgetting", since each "skill" is stored as an independent "frozen" module.

Authors test across different environments, including multi-objective tasks (where skills must be blended to meet multiple goals), settings with continual learning demands, and dynamic scenarios where the environment changes. Results show that PSEC enables efficient, flexible learning compared to vanilla RL.

**Strengths:**

The paper is easy to follow and clearly written. The experiment suite is diverse and proves the ability of SPEC to learn and compose skills.

Regarding originality and significance I particularly found interesting the usage if diffusion models to adjust the weight levels of the compositions of skills and the study for skill composiiton on the differenc spaces (parameter, noise and action spaces). It made very clear the thought process of the authors to design the framework.

**Weaknesses:**

My biggest concern with this paper is that this is not the first paper proposeing using LoRa for multiple task leanring, e.g. [1,2] and while SPEC is clearly different from previous existing approaches, some level of  comparison, theorethical or empirical would be greatly benefitial. Specifially, at present is difficult to discern what are the novel components within SPEC wrt previous skill learning frameworks.

It would be also good if authors could provide their though contrasting SPEC with existing works on RL agents that learn skills compositionally such as [3-6]. Particularly, [5] even points as an advantage over previous frameworks not having to learn and storing a different set of weight for every skill/sub-task. Since SPEC goes back to this form of learning it would be good that authors include a discussion on this topic.

[1] Ponti, Edoardo Maria, et al. "Combining parameter-efficient modules for task-level generalisation." Proceedings of the 17th Conference of the European Chapter of the Association for Computational Linguistics. 2023.

[2] Huang, Kaixin, et al. "Solving Continual Offline Reinforcement Learning with Decision Transformer." arXiv preprint arXiv:2401.08478 (2024).

[3] Araki, Brandon, et al. "The logical options framework." International Conference on Machine Learning. PMLR, 2021.

[4] Hill, F., Tieleman, O., von Glehn, T., Wong, N., Merzic, H., & Clark, S. Grounded Language Learning Fast and Slow. In ICLR 2021.

[5] León, Borja G., Murray Shanahan, and Francesco Belardinelli. "In a Nutshell, the Human Asked for This: Latent Goals for Following Temporal Specifications." ICLR 2022.

[6] Vaezipoor, Pashootan, et al. "Ltl2action: Generalizing ltl instructions for multi-task rl." International Conference on Machine Learning. PMLR, 2021.

**Questions:**

Please refer to the questions above.

--- Post discussion ---
Authors correctly addressed my concerns during the rebuttal incorporating relevant discussion on related works and limitations, while I would aprreciate some of this to be present not only in the appendix, I believe the strenghts of the work outweight the weaknesses in the revised version

---

> ### Author Response · Authors · 2024-11-20
> **Response to Reviewer ac7Q**
>
> We sincerely appreciate the reviewer for the constructive comments on our work! Regarding the concerns from the reviewer ac7Q, we provide the responses as follows:
>
> >**Weakness: It would be also good if authors could provide their though contrasting SPEC with existing works on RL agents...**
>
> - Thanks for this valuable comment! We've added discussions on all related works [1-6] mentioned by the reviewer accordingly, along with additional relevant works, in Appendix B. Here’s a brief overview of the differences:
>
>     - [1-2] use LoRA for multi-task learning but focus on static tuning, which significantly limits their flexibility and adaptability in scenarios requiring real-time compositional adjustments. In contrast, PSEC combines different LoRA via a context-aware modular, maximizing the expressiveness of pretrained skills to flexibly compose new skills. For example, in Figure 5, PSEC adaptively prioritizes safety over reward to prevent collisions.
>     - Similar to [1-2], [4] and [6] focus on static setups such as language models, whereas PSEC targets dynamic decision-making tasks, necessitating a context-aware modular for adaptive skill composition.
>     - [3] uses logical options for skill composition but requires significant human effort for skill management, limiting scalability. Additionally, [3] focuses on efficient `pretraining`, not on `fast adaptation/continual improvement`. In contrast, PSEC targets the later setups, and minimizes human effort by incorporating new skills as LoRA modules, which are then combined through auto-learned compositional networks.
>     - [5] addresses a fundamentally different problem than PSEC. So, the arguments from [5] are inapplicable here. Specifically, [5] focuses on improving model `pretraining` for enhanced OOD generalization, emphasizing the need for shared information bottlenecks across skills for strong representations. In contrast, PSEC focuses on `fast adaptation/continual improvement`, where parameter-isolated LoRA modules are essential for learning new skills, especially for preventing catastrophic forgetting.
>
> - It is difficult to ensure a fair comparison to the baselines mentioned by the reviewer for two main reasons: `different settings` and `lack of open-source code`.
>     - Specifically, [1,3-6] focus on different problems (e.g., language model, static settings) compared to the one in PSEC (continuous control, adaptive composition is crucial), making them unsuitable for comparison.
>     - The only comparable baseline that focuses on a similar task setting to PSEC is [2]. However, since its code is not publicly available, reproducing it for a fair comparison is difficult.
> - Therefore, we compare PSEC with the latest strong baseline L2M[7], which studies in a similar setting about efficient finetuning and continual learning via multi-task offline RL pretraining. Additional experiments comparing PSEC with L2M are included in the `general response` and `Appendix F` in our revised paper.
>
> Reference:
>
> [7] Schmied, Thomas, et al. "Learning to Modulate pre-trained Models in RL." Advances in Neural Information Processing Systems 36 (2024).
>
>
> We sincerely thank the reviewer for the valuable feedback. Please let us know if we have adequately addressed your concerns. If so, we would greatly appreciate it if the reviewer could consider adjusting the score.

---

> > ### Comment · Reviewer_ac7Q · 2024-11-24
> > **Follow up**
> >
> > I want to thank the authors for their response. But I would like to follow up on their thoughts you claim that your approach is better than previous works using LoRA, since authors have expressed they have no intention to provide empirical evidence of the difference between PSEC and previous works, it would require further detail in the discussion on why PSEC is so advantageous over previous work onthese scenearios. Your response is that PSEC is better because "PSEC combines different LoRA via a context-aware modular", intuitively this could be read that PSEC is better only because the modules have been extended and trained to work with different context, which is a well known practice alrady. I would like the authors to elaborate more on this.
> >
> > Regarding [5] while addresses a different problem, I grought it to showcase that raises a limitation from previous literature on skill composition that PSEC brings back again, having to train and store a library of policies and their weights that will keep growing in size and compute requirements as more skills are needed. I still don't see this limitation debated anywhere in the updated version.

---

> > > ### Author Response · Authors · 2024-11-25
> > > **Thanks for your reply! Response to follow up of reviewer ac7Q (1/3)**
> > >
> > > We really appreciate your time engaged in the review and rebuttal phase and we feel our work has been greatly improved through your constructive comments! Regarding the remaining concerns, we provide the following detailed discussions and experiments:
> > >
> > > > **Advantages over previous methods**
> > >
> > > We acknowledge that both [1] and [2] adopt LoRA to encode different skills like us. However, PSEC primarily differs fundamentally in two different key axes: `How to pretrain different skills`, and `How to leverage previous knowledge to adapt to new skills`.
> > >
> > > - `How to pretrain different skills?`
> > >     - Specifically, [1] proposes training all LoRA modules jointly during pretraining, with tasks mapped to a fixed set of discrete latent skills via a predefined task–skill allocation matrix, suffering from the following limitations:
> > >         1.	Fixed Skill Inventory: A predefined skill set limits flexibility. Its size, determined via prior validation, may fail to represent new tasks, reducing expressiveness.
> > >         2.	Complex Regularization: It requires additional regularizations such as IBR prior to avoid entropic and general-purpose modules, and sparse allocation metrics, which increases complexity and computational cost.
> > >     - [2] requires training a separate Decision Transformer (DT) with **FULL** parameter size alongside the LoRAs for fine-tuning new tasks. This reliance on additional models introduces several limitations:
> > >         1.	Increased Data Requirements: Training a separate DT model demands significant data. Limited data can lead to suboptimal skill learning.
> > >         2.	Suboptimal Skill Generalization: Insufficient data may impair the model’s ability to capture diverse skill representations, reducing performance across tasks.
> > >     - PSEC trains each LoRA module independently, eliminating parameter conflicts associated with learning multiple skills simultaneously. This separation allows task-specific knowledge to be captured effectively without cross-skill interference. Furthermore, PSEC avoids the need for training additional large-scale models for new tasks, focusing solely on LoRA modules tailored to individual skills. This lightweight design ensures efficient skill-specific adaptation, reducing computational overhead while preserving adaptability and performance across diverse tasks.

---

> ### Author Response · Authors · 2024-11-25
> **Response to follow up of reviewer ac7Q (2/3)**
>
> - `How to leverage previous knowledge to adapt to new skills?`
>   - [1] directly initialize from the pretrained LoRA and finetune it.
>     - `Limitation`: It will cause catastrophic forgetting. After finetuning on new tasks, the performance on the previous tasks may be dropped.
>   - [2] manually merges different skills before fine-tuning LoRA.
>     -  `Limitations`:
>
>          1.Manual Weight Dependency: Task performance depends on manually determined weights between the previous model and the current task model. This can lead to unstable training if the merged parameters are hard to manage.
>
>          2.Suboptimal and Unstable Skill Composition: Manually composing skills may fail to leverage shared structures across skills, potentially overwriting learned knowledge and resulting in suboptimal performance, especially when some skills are suboptimal caused by limited data.
>   - PSEC offers flexibility in adapting to various settings for new tasks.
>       1. Composition Stability: PSEC can avoid catastrophic forgetting by adhering a parameter-isolation manner.
>       2. Composition Optimality: PSEC avoids manually composing via a context-aware modular optimized via gradient descent. This ensures efficient skill integration and preserves task-specific knowledge.
>
> - To demonstrate this, we provide more empirical evidence by evaluating the MetaWorld benchmark (18 pretrained tasks and 12 unseen tasks):
>
> Setting:
> We implemented skill modeling as described in [1], setting the skill inventory size to 18, matching the pretraining tasks. The model is pretrained on datasets from 18 Meta-World tasks, `jointly` optimizing the skill allocation matrix and skill inventories, with each skill represented as a LoRA module. The evaluation results for these 18 pretrained tasks and an additional set of 12 unseen tasks in the `zero-shot` setting are summarized in the following table.
>
> ### 18 pretrained tasks
> |Tasks| method in [1]  | PSEC     |
> |---- | ---- | ---|
> |peg-insert-side-v2| 0.88 | **0.90** |
> |peg-unplug-side-v2| 0.56 | **0.86** |
> |button-press-topdown-v2 | 0.85 | **0.89** |
> |push-back-v2| 0.0  | **0.88** |
> |window-close-v2  | 0.39 | **0.88** |
> |door-open-v2| 0.85 | **0.86** |
> |handle-press-v2  | 0.96 | **0.97** |
> |plate-slide-side-v2| 0.35 | **0.74** |
> |handle-pull-side-v2| 0.94 | **0.95** |
> |window-open-v2| 0.73 | **0.89** |
> |door-close-v2| 0.90 | **0.91** |
> |reach-v2| 0.95 | **0.95** |
> |push-v2 |0.60 | **0.92** |
> |stick-push-v2  | 0.0| **0.79** |
> |drawer-close-v2 | 0.90 | **0.97** |
> |plate-slide-back-v2| 0.93 | **0.95** |
> |coffee-button-v2| 0.91 | **0.95** |
> |hand-insert-v2 | 0.34 | **0.89** |
> |Mean| 0.67 |**0.90** |
>
> ### 12 zero-shot tasks
> | Tasks  | method in [1]  | PSEC|
> |---|----|---|
> |plate-slide-v2 | 0.0| **0.15** |
> |handle-press-side-v2| 0.50 | **0.62** |
> |button-press-wall-v2| 0.0| **0.40** |
> |button-press-topdown-wall-v2 | 0.86 | **0.89** |
> |push-wall-v2 |0.33| **0.71** |
> |reach-wall-v2 |0.58| **0.90** |
> |faucet-close-v2|0.0| **0.16** |
> |button-press-v2|0.0| **0.15** |
> |plate-slide-back-side-v2| 0.0 | 0.0 |
> |handle-pull-v2| 0.0| 0.0 |
> |faucet-open-v2| 0.0| **0.77** |
> |stick-pull-v2|0.0 | 0.0 |
>
> We also compare PSEC against the method described in [2]. The method in [2] involves training an additional model with **FULL** parameter for each task and manually merging all parameters beyond the LoRA components. We evaluate it on the 12 tasks in Meta-World with `few-shot` setting.
>
> ### 12 few-shot tasks
> | Tasks | method in [2] | PSEC     |
> | ---- | ---- | ---- |
> | plate-slide-v2 | 0.60| **0.89** |
> | handle-press-side-v2 | 0.72| **0.92** |
> | button-press-wall-v2| 0.09| **0.72** |
> | button-press-topdown-wall-v2 | 0.04| **0.89** |
> | push-wall-v2| 0.02| **0.88** |
> | reach-wall-v2   | 0.21| **0.90** |
> | faucet-close-v2| 0.76| **0.90** |
> | button-press-v2  | 0.18| **0.23** |
> | plate-slide-back-side-v2| 0.08| **0.92** |
> | handle-pull-v2| 0.10| **0.93** |
> | faucet-open-v2 | 0.17| **0.89** |
> | stick-pull-v2 | 0   | **0.32** |
>
> ## Analysis:
> The results demonstrate PSEC’s superior performance over [1] and [2] across 18 tasks and 12 unseen tasks. Specifically:
>
> - Zero-shot Setting: [1] suffers from its potentially entropic latent skills, which struggle to generalize to unseen tasks. Also, the inferior results on the 18 tasks can be attributed to parameter conflicts from training all LoRA models together, destabilizing learning.
>
> - Few-shot Setting: PSEC outperforms [2] on the 12 unseen tasks. The additional model in [2] needs to tune far more parameters than PSEC and thus faces difficulties in the few-shot setting, leading to suboptimal performance. Furthermore, the manual merging of skills in [2] fails to leverage shared structures across skills and risks at overwriting previously learned knowledge.
>
> These results highlight PSEC’s efficiency and robustness in utilizing prior knowledge, avoiding parameter conflicts, and maintaining strong performance in both zero-shot and few-shot settings compared to previous methods.

---

> > ### Author Response · Authors · 2024-11-25
> > **Response to follow up of reviewer ac7Q (3/3)**
> >
> > > **Linearly growth computational cost**
> >
> > - Thanks for this valuable feedback and sorry for the confusion! Actually, the limitation mentioned by the reviewer is discussed in `Appendix A (Redundant skill expansion)`, which could be potentially addressed by employing an evaluation metric to assess the interconnections between skills, helping determine when to expand the skill library.
> >     - Also, we believe the linear growth in computation and memory usage is acceptable, given its potential to exponentially increase expressiveness by incorporating more skills. For example, $2^n=C_n^0+C_n^1+C_n^2+...+C_n^n$ skills could be composed of n different (non-redundant) skills by using binary compositional weight (0 for deactivate and 1 for activate). In our work, we adopt a more expressive context-aware compositional method, enabling the formation of even more complex behaviors from these skill primitives.
> >
> >
> > - In addition, we want to discuss further the advantages of modularized training over the one mentioned in [5].
> >
> >     - **Easy to expand**. First, we want to acknowledge again that [5] and PSEC study different problems, so they are `orthogonal` rather than `contradictory`. Specifically, adopting a shared representation in [5] is beneficial for `pretraining` by extracting generalizable features across diverse skills, but it risks `catastrophic forgetting during continual learning` in PSEC's context.
> >         - We agree with [5] on the value of a shared structure for pretraining. So, we also adopt a $\pi_0$ pretrained on diverse distributions, as discussed in Appendix A (Assumption on the expressiveness of the pretrain policy).
> >         - However, finetuning this shared feature can lead to catastrophic forgetting during fine-tuning, if the pretrained knowledge is not well preserved or regularized. In contrast, PSEC’s modularized approach avoids this by encoding new skills in separate LoRA modules without altering pretrained ones, using a parameter-isolation strategy, which could also enhance [5] by mitigating catastrophic forgetting during fine-tuning.
> >
> >     - **Easy to manage and compose**. By adopting a modularized design, PSEC can conveniently manage its skill library by discarding the bad ones while retaining the good ones, ensuring scalability and manageability when incorporating a large skill corpus, as discussed in `Appendix B line917~921`. [5], however, cannot achieve this because it is hard to modify the shared features after pretraining and ensure stability during finetuning.
> >
> >
> > Thanks for your time and valuable feedback again! Please feel free to let us know if the reviewer has further comments. It would be really appreciated if the reviewer could consider re-evaluating our paper.

---

> > > ### Comment · Reviewer_ac7Q · 2024-12-01
> > > **Follow up 2**
> > >
> > > I want to thank the authors for all the additional details and addressing my remaining concerns! Yes this response and all the details mentioned address my concerns on comparison with the existing literature and limitations of the work. I am updating my evaluation accordingly to recommend acceptance

---

> > > > ### Author Response · Authors · 2024-12-02
> > > > **Thanks for increasing the score!**
> > > >
> > > > Thanks for your reply and for raising your score! It's happy to hear that we've successfully addressed your concerns!

---

### Author Response · Authors · 2024-11-20
**General Response (1/2)**

We sincerely thank all reviewers for their valuable feedback, which greatly enhances the quality of our paper. Regarding their concerns, we've provided detailed rebuttals. Here, we summarize our paper revisions and introduce the new experiments on the Meta-World benchmark. Now, we've extensively evaluated the advantages of PSEC on `4 benchmarks` including DSRL, DMC Control Suite, D4RL, and Meta-World. Hope our rebuttal can properly address the concerns, and we would be no more happy to hear any further comments that can improve our paper.

>**1. Paper revision summarization**.

All revisions are marked as red text color.
- (For reviewer 18jx) We revised the confusing terms such as 'rapid', 'versatility' and 'safety'.
- (For reviewer 18jx) We carefully revised the paper to address grammatical issues and will continue to proofread it to minimize such errors.
- (For reviewer ac7Q, 18jx) We added the discussions on more relevant works in Appendix B.
- (For reviewer 18jx, oGS2) We added detailed descriptions of the safe offline RL benchmark in Appendix E.
- (For reviewer 18jx, 1bSW, oGS2) We added more complex experiments covering 50 skills on Meta-World in Appendix F.
- (For reviewer 18jx) We added a new visualization to support the advantages of PSEC in Appendix G.


>**2. Evaluation on more complex benchmarks that cover more skills**.

To evaluate the effectiveness of PSEC on more complex experiments, we conduct the experiments on Meta-World benchmark, which consists of 50 diverse tasks for robotic manipulation, such as grasping, manipulating objects, opening/closing a window, pushing buttons, locking/unlocking a door, and throwing a basketball. We compare PSEC with the strong baseline L2M[2]. The experiments include three experiment settings.

* `Continual Learning Setting`. We follow Continual world[1], L2M[2] and split the 50 tasks into 40 pre-training tasks, and 10 fine-tuning unseen tasks (CW10). The training datasets are the same as the datasets collected by L2M[2]. We train 10K steps per task in CW10, which is only 10% steps of L2M, with a batch size of 1024. After every 10K update steps, we switch to the next task in the sequence. Then we evaluate it on all tasks in the task sequence. The results are shown in the following Table. We compare the performance of PSEC with L2M and other baselines. The results demonstrate that PSEC can achieve better performance on complex tasks.

| Methods    | Success rate |
| ---------- | ------------ |
| L2M        | 0.65         |
| L2M-oracle | 0.77         |
| L2P-Pv2    | 0.4          |
| L2P-PreT   | 0.34         |
| L2P-PT     | 0.23         |
| EWC        | 0.17         |
| L2         | 0.1          |
| PSEC       | **0.87**       |

---

> ### Author Response · Authors · 2024-11-20
> **General Response (2/2)**
>
> * `Unseen Tasks Setting`. To further evaluate the efficiency of PSEC on more challenging tasks, we pretrain on fewer (18) tasks and evaluate it on more (12) unseen tasks than the first setting. Firstly, we pretrain and finetune on 18 tasks to obtain 18 LoRA modules. The performance on the 18 pretrained tasks is shown in the following Table. Results show that PSEC can achieve enhanced skill learning even when the pretrained model is combined with one LoRA for each task if the skill is composed in parameter space.
>
> | Tasks                   | Scratch | ASEC | NSEC | PSEC |
> | ----------------------- | ------- | ---- | ---- | ---- |
> | peg-insert-side-v2      | 0.50    | 0.87 | 0.88 | **0.90** |
> | peg-unplug-side-v2      | 0.35    | 0.61 | 0.78 | **0.86** |
> | button-press-topdown-v2 | 0.71    | 0.88 | 0.88 | **0.89** |
> | push-back-v2            | 0.26    | 0.61 | 0.76 | **0.88** |
> | window-close-v2         | 0.65    | 0.84 | 0.84 | **0.88** |
> | door-open-v2            | 0.74    | 0.85 | **0.86** | **0.86** |
> | handle-press-v2         | 0.67    | 0.96 | **0.97** | **0.97** |
> | plate-slide-side-v2     | 0.27    | 0.23 | 0.53 | **0.74** |
> | handle-pull-side-v2     | 0.76    | 0.94 | 0.94 | **0.95** |
> | window-open-v2          | 0.87    | 0.75 | 0.88 | **0.89** |
> | door-close-v2           | 0.90    | 0.89 | 0.89 | **0.91** |
> | reach-v2                | 0.89    | **0.95** | **0.95** | **0.95** |
> | push-v2                 | 0.15    | 0.58 | 0.81 | **0.92** |
> | stick-push-v2           | 0.44    | 0.54 | 0.17 | **0.79** |
> | drawer-close-v2         | **0.97**    | **0.97** | **0.97** | **0.97** |
> | plate-slide-back-v2     | 0.90    | 0.94 | 0.94 | **0.95** |
> | coffee-button-v2        | 0.91    | 0.94 | 0.94 | **0.95** |
> | hand-insert-v2          | 0.30    | 0.68 | 0.63 | **0.89** |
> | Mean  |  0.62       |   0.78   |   0.81   |  0.90    |
>
> * Then, we evaluate PSEC with the obtained 18 LoRA modules on the unseen tasks. For the unseen tasks, we conduct two types of experiments: few-shot setting and zero-shot setting.
>
> `Few-shot` We perform few-shot learning by training the context-aware module for 1k steps using only 10% of the total data available for unseen tasks. This setup simulates scenarios with limited data on new tasks. The results, summarized in the following Table, demonstrate that PSEC achieves a high success rate on unseen tasks. This indicates that PSEC can effectively adapt to new tasks, showcasing its capability for rapid transfer learning and efficient adaptation in data-scarce environments.
>
> | Tasks | ASEC    | NSEC    | PSEC |
> | ----- | --- | --- | ---- |
> | plate-slide-v2| 0.14    | 0.66    | **0.89** |
> | handle-press-side-v2| 0.73    | 0.65    | **0.92** |
> | button-press-wall-v2 | 0.09    |0.03     | **0.72** |
> | button-press-topdown-wall-v2 | 0.87    | 0.88    | **0.89** |
> | push-wall-v2 | 0.57    | 0.68    | **0.88** |
> | reach-wall-v2 | 0.41    | 0.36    | **0.90** |
> | faucet-close-v2  | 0.41    | 0.49    | **0.90** |
> | button-press-v2 | 0.02    | 0.14    | **0.23** |
> | plate-slide-back-side-v2 | 0.17    | 0.19    | **0.92** |
> | handle-pull-v2   | 0.15    | 0.21    | **0.93** |
> | faucet-open-v2  | 0.14    | 0.16    | **0.89** |
> | stick-pull-v2   | 0    | 0    | **0.32** |
>
> `Zero-shot` In this setting, no data from the unseen tasks is used to train the context-aware compositional modular. Instead, this module is trained for 2k steps using datasets from 18 pre-trained tasks. It is then evaluated directly on 12 unseen tasks. Interestingly, even without access to any data from the unseen tasks, PSEC demonstrates strong performance on several tasks. Notably, PSEC substantially outperforms NSEC and ASEC on this zero-shot transfer setting, highlighting the advantages of skill compositions in parameter spaces over noise and action spaces. Overall, the results demonstrate PSEC’s ability to effectively utilize knowledge from previously learned skills to achieve strong zero-shot transfer.
>
> | Tasks| ASEC | NSEC | PSEC |
> | ---------------------------- | ---- | ---- | ---- |
> | plate-slide-v2 | 0.03 |  0.0    | **0.15** |
> | handle-press-side-v2| 0.50 | 0.60     | **0.62** |
> | button-press-wall-v2 | 0.0  | 0.0     | **0.40** |
> | button-press-topdown-wall-v2 | 0.85 | 0.87     | **0.89** |
> | push-wall-v2| 0.53 | 0.53     | **0.71** |
> | reach-wall-v2  | 0.34 | 0.05     | **0.90** |
> | faucet-close-v2   | 0.0  | 0.0     | **0.16** |
> | button-press-v2  | 0.0  | 0.0     | **0.15** |
> | plate-slide-back-side-v2     | 0.0  | 0.0     | 0.0  |
> | handle-pull-v2| 0.0  | 0.0     | 0.0  |
> | faucet-open-v2  | 0.0  | 0.0     | **0.77** |
> | stick-pull-v2 | 0.0  | 0.0     | 0.0  |
>
> Reference:
>
> [1] Wolczyk, M., Zajkac, M., Pascanu, R., Kucin ́ski, L., and Milos ́, P.. Continual world: A robotic benchmark for continual reinforcement learning. Advances in Neural Information Processing Systems (2021)
>
> [2] Schmied, Thomas, et al. "Learning to Modulate pre-trained Models in RL." Advances in Neural Information Processing Systems 36 (2024).

---

### Meta-Review · Area_Chair_sY48 · 2024-12-18

**Metareview:**

The paper presents an approach for learning skills and adapting them offline datasets. The paper is quite interesting with all the reviewers agreeing on the novelty of the work. Particularly the idea of using LORA to adapt the agent's skills is exciting. There were minor concerns on clarity, algorithmic details, and grammatical errors. Many of them have been addressed in the rebuttal phase. Overall, this is a solid paper that can have significant impact in the RL community.

**Additional Comments On Reviewer Discussion:**

After discussion, several of the minor concerns on paper clarity and lack of details were resolved.

---

### Decision · Program_Chairs · 2025-01-22

Accept (Poster)